# Batch Active Learning at Scale

**Gui Citovsky, Giulia DeSalvo, Claudio Gentile, Lazaros Karydas,**
**Anand Rajagopalan, Afshin Rostamizadeh, Sanjiv Kumar**
Google Research
{gcitovsky,giuliad,cgentile,lkary,anandbr,rostami,sanjivk}@google.com

## Abstract

The ability to train complex and highly effective models often requires an abundance of training data, which can easily become a bottleneck in cost, time, and computational resources. Batch active learning, which adaptively issues batched queries to a labeling oracle, is a common approach for addressing this problem. The practical benefits of batch sampling come with the downside of less adaptivity and the risk of sampling redundant examples within a batch – a risk that grows with the batch size. In this work, we analyze an efficient active learning algorithm, which focuses on the large batch setting. In particular, we show that our sampling method, which combines notions of uncertainty and diversity, easily scales to batch sizes (100K-1M) several orders of magnitude larger than used in previous studies and provides significant improvements in model training efficiency compared to recent baselines. Finally, we provide an initial theoretical analysis, proving label complexity guarantees for a related sampling method, which we show is approximately equivalent to our sampling method in specific settings.

## 1 Introduction

Training highly effective models for complex tasks often hinges on the abundance of training data. Acquiring this data can easily become a bottleneck in cost, time, and computational resources. One major approach for addressing this problem is active learning, where labels for training examples are sampled selectively and adaptively to more efficiently train the desired model over several iterations. The adaptive nature of active learning algorithms, which allows for improved data-efficiency, comes at the cost of frequent retraining of the model and calling the labeling oracle. Both of these costs can be significant. For example, many modern deep networks can take days or weeks to train and require hundreds of CPU/GPU hours. At the same time, training human labelers to become proficient in potentially nuanced labeling tasks require significant investment from both the designers of the labeling task and the raters themselves. A sufficiently large set of queries should be queued in order to justify these costs.

To address these overhead costs, previous works have developed algorithms for the *batch active learning* setting, where label requests are batched and model updates are made less frequently, reducing the number of active learning iterations. Of course, there is a trade-off, and the practical benefits of batch sampling come with the downside of less adaptivity and the risk of sampling redundant or otherwise less effective training examples within a batch. Batch active learning methods directly combat these risks in several different ways, for example, by incorporating diversity inducing regularizers or explicitly optimizing over the choice of samples within a batch to optimize some notion of information.

However, as the size of datasets grows to include hundreds of thousands and even millions of labeled examples (cf. Deng et al. [2009], Krasin et al. [2017], Van Horn et al. [2018]), we expect the active learning batch sizes to grow accordingly as well. The challenge with very large batch sizes is two-fold: first, the risks associated with reduced adaptivity continue to be compounded and, second, the batch

35th Conference on Neural Information Processing Systems (NeurIPS 2021).

sampling algorithm must scale well with the batch size and not become a computational bottleneck itself. While previous works have evaluated batch active learning algorithms with batch-sizes of thousands of points (e.g., Ash et al. [2020], Sener and Savarese [2018]), in this work, we consider the challenge of active learning with batch sizes one to two orders of magnitude larger.

In this paper, we develop, analyze, and evaluate a batch active learning algorithm called Cluster-Margin, which we show can scale to batch sizes of 100K or even 1M while still providing significantly increased label efficiency. The main idea behind Cluster-Margin is to leverage Hierarchical Agglomerative Clustering (HAC) to diversify batches of examples that the model is least confident on. A key benefit of this algorithm is that HAC is executed only once on the unlabeled pool of data as a preprocessing step for all the sampling iterations. At each sampling iteration, this algorithm then retrieves the clusters from HAC over a set of least confident examples and uses a round-robin scheme to sample over the clusters.

The contributions of this paper are as follows:

- We develop a novel active learning algorithm, Cluster-Margin, tailored to large batch sizes that are orders of magnitude larger than what have been considered in the literature.
- We conduct large scale experiments using a ResNet-101 model applied to multi-label Open Images Dataset consisting of almost 10M images and 60M labels over 20K classes, to demonstrate significant improvement Cluster-Margin provides over the baselines. In the best result, we find that Cluster-Margin requires only 40% of the labels needed by the next best method to achieve the same target performance.
- To compare against latest published results, we follow their experimental settings and conduct smaller scale experiments using a VGG16 model on multiclass CIFAR10, CIFAR100, and SVHN datasets, and show Cluster-Margin algorithm's competitive performance.
- We provide an initial theoretical analysis, proving label complexity guarantees for a margin-based clustering sampler, which we then show is approximately equivalent to the Cluster-Margin algorithm in specific settings.

## 1.1 Related Work

The remarkable progress in Deep Neural Network (DNN) design and deployment at scale has seen a vigorous resurgence of interest in active learning-based data acquisition for training. Among the many active learning protocols available in the literature (pool-based, stream-based, membership query-based, etc.), the batch pool-based model of active learning has received the biggest attention in connection to DNN training. This is mainly due to the fact that this learning protocol corresponds to the way labels are gathered in practical large-scale data processing pipelines.

Even restricting to batch pool-based active learning, the recent literature has become quite voluminous, and we can hardly do it justice here. In what follows, we briefly mention what we believe are among the most relevant papers to our work, with a special attention to scalable methods for DNN training that delivered state of the art results in recently reported experiments.

In Sener and Savarese [2018], the authors propose a CoreSet approach to enforce diversity of sampled labels on the unlabeled batch. There, the CoreSet idea was used as a way to compress the batch into a subset of representative points. No explicit notion of informativeness of the data in the batch is adopted. The authors reported an interesting experimental comparison on small-sized datasets. Yet, their Mixed Integer Programming approach to computing CoreSets becomes largely infeasible as the batch size grows and the authors suggest a 2-approximation algorithm as a solution. As we find empirically, it seems this lack of an informativeness signal, perhaps coupled with the 2-approximation, limits the effectiveness of the method in the large batch-size regime.

Among the relevant papers in uncertainty sampling for batch active learning is Kirsch et al. [2019], where the uncertainty is provided by the posterior over the model weights, and diversity over the batch is quantified by the mutual information between the batch of points and model parameters. Yet, for large batch sizes and standard acquisition functions, their method also becomes infeasible in practice (we discuss this in more detail in the Section 3).[1] Another relevant recent work, and one

---

[1]More recently, i.e., contemporaneously with this publication, a more efficient variant of BatchBALD has been proposed by Kirsch et al. [2021]. The authors propose a simple idea to turn the original BALD algorithm of Houlsby et al. [2011] into a batch active learning algorithm via a softmax function over the current uncertainties

---

**Algorithm 1** Hierarchical Agglomerative Clustering (HAC) with Average-Linkage

---

**Require:** Set of clusters $C$, distance threshold $\epsilon$, minimum cluster distance $d = min_{A \neq B \in C} d(A, B)$.

1: **if** $|\mathcal{C}| = 1$ **or** $d > \epsilon$ **then**
2:     **return** $\mathcal{C}$.
3: **end if**
4: $A, B \leftarrow$ Pair of distinct clusters in $\mathcal{C}$ which minimize $d(A, B) = \frac{1}{|A||B|} \sum_{a \in A, b \in B} d(a, b)$.
5: **if** $d(A, B) \leq \epsilon$ **then**
6:     $\mathcal{C} \leftarrow \{A \cup B\} \cup \mathcal{C} \setminus \{A, B\}$.
7: **end if**
8: **return** $\text{HAC}(\mathcal{C}, \epsilon, d(A, B))$.

---

which we will compare to, is Ash et al. [2020], where a sampling strategy for DNNs is proposed which uses $k$-MEANS++ seeding on the gradients of the final layer of the network in order to query labels that balance uncertainty and diversity. One potential downside to this approach is that the dimension of the gradient vector grows with the number of classes. In Wei et al. [2015] (see also the more recent Killamsetty et al. [2020]), the authors propose a submodular sampling objective that trades-off model uncertainty with a diversity-inducing regularizer, such as a facility location objective. Using naive greedy optimization to solve such objectives does not immediately scale to extremely large batch sizes of hundreds of thousands or more (such an implementation would require a linear number of function evaluations per greedy example added to the batch). More efficient "lazier-than-lazy" stochastic approximations (e.g., Mirzasoleiman et al. [2015]) may be able to scale to very large batch sizes as they require only a linear number of function evaluations overall (modulo a $\log(1/\epsilon)$ factor, where $\epsilon$ is the approximation parameter). However, this may still be impractical if computing the marginal gain is expensive (for example, if it grows approximately linearly with the pool size).

Further recent works related to DNN training through batch active learning are Zhdanov [2019], Shui et al. [2020], Kim et al. [2020], Ghorbani et al. [2021]. In Zhdanov [2019], the authors trade off informativeness and diversity by adding weights to $k$-means clustering. The idea is similar in spirit to our proposed algorithm, though the way we sample within the clusters is very different (see Section 2 below). Shui et al. [2020] proposes a unified method for both label sampling and training, and indicates an explicit informativeness-diversity trade-off in label selection. The authors model the interactive procedure in active learning as a distribution matching problem measured by the Wasserstein distance. The resulting training process gets decomposed into optimizing DNN parameters and batch query selection via alternating optimization. We note that such modifications in the training procedure are not feasible in settings where only data selection can be modified while the training routine is treated as a black-box (very frequent in practice). The work of Kim et al. [2020] is based on the idea that uncertainty-based methods do not fully leverage the data distribution, while data distribution-based methods often ignore the structure of the learning task. Hence the authors propose to combine them in Variational Adversarial Active Learning method from Sinha et al. [2019], the loss prediction module from Yoo and Kweon [2019], and RankCGAN from Saquil et al. [2018]. Despite the good performance reported on small datasets, these techniques are not geared towards handling large batch sizes, which is the goal of our work.

From this lengthy literature, we focus on the BADGE [Ash et al., 2020] and CoreSet algorithms [Sener and Savarese, 2018] (described in more detail in Section 3) as representative baselines to compare against since they are relatively scalable in terms of the batch size, do not require modification of the model training procedure, and have shown state-of-the-art results on several benchmarks.

## 2 Algorithm

In this section, we present the Cluster-Margin algorithm, whose pseudo-code is in Algorithm 2. Throughout, we refer to the model being trained as $f$ and denote its corresponding weights/parameters $w$. Given an unlabeled pool of examples $X$, in each sampling iteration, Cluster-Margin selects a diverse set of examples on which the model is least confident. We compute the confidence of a model

---

in the pool. This idea yields a faster and simpler algorithm than BatchBALD, but it does not appear to explicitly model diversity, other than sampling without replacement from the corresponding Gibbs distribution.

**Algorithm 2** The Cluster-Margin Algorithm

---

**Require:** Unlabeled pool $X$, neural network $f$, seed set size $p$, number of labeling iterations $r$, margin batch size $k_m$, target batch size $k_t \leq k_m$, HAC distance threshold $\epsilon$.

1: $S \leftarrow \emptyset$, the set of labeled examples.
2: Draw $P \subset X$ ($|P| = p$) seed set examples uniformly at random and request their labels. Set $S \leftarrow S \cup P$.
3: Train $f$ on $P$.
4: Compute embeddings $E_X$ on the entire set $X$, using the penultimate layer of $f$.
5: $\mathcal{C}_X \leftarrow \text{HAC}(E_X, \epsilon, 0)$.
6: **for** $i = 1, 2, \ldots, r$ **do**
7:     $S_i \leftarrow \emptyset$.
8:     $M_i \leftarrow$ The $k_m$ examples in $X \setminus S$ with smallest margin scores.
9:     $\mathcal{C}_{M_i} \leftarrow$ Mapping of $M_i$ onto $\mathcal{C}_X$.
10:     Sort $\mathcal{C}_{M_i}$ ascendingly by cluster size. Set $\mathcal{C}'_{M_i} \leftarrow [C_1, C_2, \ldots, C_{|\mathcal{C}_{M_i}|}]$ as the sorted array, and set $j \leftarrow 1$ as the index into $\mathcal{C}'_{M_i}$.
11:     **while** $|S_i| < k_t$ **do**
12:         Select $x$, a random example in $C_j$.
13:         $S_i \leftarrow S_i \cup \{x\}$.
14:         **if** $j < |\mathcal{C}'_{M_i}|$ **then**
15:             $j \leftarrow j + 1$.
16:         **else**
17:             $j \leftarrow$ The index of the smallest unsaturated cluster in $\mathcal{C}'_{M_i}$.
18:         **end if**
19:     **end while**
20:     Request labels for $S_i$ and set $S \leftarrow S \cup S_i$.
21:     Train $f$ on $S$.
22: **end for**
23: **return** $S$.

---

on an example as the difference between the largest two predicted class probabilities, just as is done in the so-called "margin" uncertainty sampling variant [Roth and Small, 2006], and refer to the value as the *margin score*. The lower the margin score, the less confident the model is on a given example. In order to ensure diversity among the least confident examples, we cluster them using HAC with average-linkage (Algorithm 1). The batch of examples for which labels are requested is then chosen such that each cluster is represented in the batch.

In the following, we describe the Cluster-Margin algorithm in detail, which can be decomposed into an initialization step, a clustering step, and a sequence of sampling steps.

**Initialization step.** Cluster-Margin starts by selecting a seed set $P$ of examples uniformly at random, for which labels are requested. A neural network is trained on $P$ and each $x \in X$ is then embedded with the penultimate layer of the network.

**Clustering step.** HAC with average-linkage is run as a preprocessing step. Specifically, Algorithm 1 is run *once* on the embeddings of the examples in the *entire* pool, $X$, to generate clusters $\mathcal{C}$. As described in Algorithm 1, HAC with average-linkage repeatedly merges the nearest two clusters $A, B$, so long as the distance $d(A, B) = \frac{1}{|A||B|} \sum_{a \in A, b \in B} d(a, b) \leq \epsilon$, where $\epsilon$ is a predefined threshold. In order to achieve speedup in this step, we describe an approach in Appendix A.2, where HAC is run only on the initial seed set $P$, and the examples in $X \setminus P$ are projected onto the clusters of $P$. This reduces the clustering run-time from $O(|X|^2 \log |X|)$ to $O(|P|^2 \log |P| + |P||X \setminus P|)$, without any observed loss in performance in our experiments.

**Sampling steps.** In each labeling iteration, we employ a sampling step that selects a diverse set of low confidence examples. This process first selects $M$, a set of unlabeled examples with lowest margin scores, with $|M| = k_m$. Then, $M$ is filtered down to a diverse set of $k_t$ examples, where $k_t$ is the target batch size per iteration ($k_t \leq k_m$). Specifically, given the $k_m$ lowest margin score examples, we retrieve their clusters, $\mathcal{C}_M$, from the clustering step and then perform the diversification process. To select a diverse set of examples from $\mathcal{C}_M$, we first sort $\mathcal{C}_M$ ascendingly by cluster size. As depicted in Figure 1, we then employ a round-robin sampling scheme whereby we iterate through

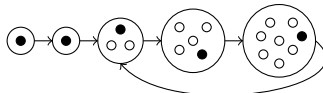

Figure 1: Random round-robin sampling from clusters.

the clusters in the sorted order, selecting one example at random from each cluster, and returning to the smallest unsaturated cluster once we have sampled from the largest cluster. This process is then repeated until $k_t$ examples have been selected. We sample from the smallest clusters first as they come from the sparsest areas of the embedded distribution and contain some of the most diverse points. We then leverage round-robin sampling to maximize the number of clusters represented in our final sample.

Cluster-Margin is simple to implement and admits the key property of only having to run the clustering step once as preprocessing. As we will see, this is in contrast to recent active learning algorithms, such as BADGE and CoreSet, that run a diversification algorithm at each sampling iteration. HAC with average-linkage runs in time $O(n^2 \log n)$ where $n = |X|$, but lends itself to massive speedup in practice from multi-threaded implementations [Sumengen et al., 2021]. At each iteration, Cluster-Margin takes time $O(n \log n)$ to sample examples, whereas BADGE and CoreSet take time $O(k_t n)$, which in the large batch size setting (e.g., $k_t = \Omega(\sqrt{n})$) can be far more expensive in practice.

## 3 Empirical Evaluation

Here, we present thorough experimental results comparing the Cluster-Margin algorithm against several state-of-the-art baselines on different image datasets. We consider both the large scale Open Images dataset and small scale datasets including CIFAR10, CIFAR100, SVHN. Depending on the type of dataset, we consider different neural network architectures, which we describe in detail below. Finally, additional baselines and/or data sets that have been added to the initial version of this article are presented in Appendix A.3. This appendix will continue to be updated as further evaluations are completed and the most up-to-date version will be found at https://arxiv.org/abs/2107.14263.

### 3.1 Baselines Considered

For all experiments, we consider a set of baselines that consists of a classical Uncertainty Sampling algorithm, as well as two recent active learning algorithms, BADGE and CoreSet, that have been shown to work well in practice [Ash et al., 2020, Sener and Savarese, 2018]. We also conducted initial comparisons to the FASS algorithm of Wei et al. [2015] (see Appendix A.3).

**Uncertainty Sampling (Margin Sampling):** selects $k$ examples with the smallest model confidence (or highest uncertainty) as defined by the difference of the model's class probabilities of the first and second most probable classes. That is, letting $X$ be the current unlabeled set, Uncertainty Sampling selects examples $x \in X$ that attain $\min_{x \in X} \Pr_w[\hat{y}_1 | x] - \Pr_w[\hat{y}_2 | x]$ where $\Pr_w[\hat{y}|x]$ denotes the probability of class label $\hat{y}$ according the model weights $w$ and where $\hat{y}_1 = \arg\max_{y \in \mathcal{Y}} \Pr_w[y|x]$ and $\hat{y}_2 = \arg\max_{y \in \mathcal{Y}/\hat{y}_1} \Pr_w[y|x]$ are the first and second most probable class labels according to the model $w$ [Roth and Small, 2006].

**BADGE:** selects $k$ examples by using the $k$-MEANS++ seeding algorithm on $\{g_x : x \in X\}$ where $g_x$ is the gradient embedding of example $x$ using the current model weights $w$. For cross-entropy loss and letting $|\mathcal{Y}| = l$ denote the number of classes, the gradient $g_x = [g_x(1), ..., g_x(l)]$ is composed of $l$ blocks. For each $y \in [l]$, $g_x(y) = (\Pr_w[y|x] - 1_{\hat{y}_1 = y})z_x$ where $z_x$ is the penultimate embedding layer of the model on example $x$ and $\hat{y}_1 = \arg\max_{y \in \mathcal{Y}} \Pr_w[y|x]$ is the most probable class according to the model weights $w$ [Ash et al., 2020].

**Approximate CoreSet ($k$-center):** selects $k$ examples by solving the $k$-center problem on $\{z_x : x \in X\}$, where $z_x$ is the embedding of $x$ derived from the penultimate layer of the model [Sener and Savarese, 2018]. In the original algorithm of Sener and Savarese [2018], the authors solve a mixed integer program based on $k$-center. Instead, we use the classical greedy 2-approximation where the next center is chosen as the point that maximizes the minimum distance to all previously chosen centers, which is also suggested by the authors when greater computational efficiency is required.

|            | Images    | Positives  | Negatives  |
|------------|-----------|------------|------------|
| Train      | 9,011,219 | 19,856,086 | 37,668,266 |
| Validation | 41,620    | 367,263    | 228,076    |
| Test       | 125,436   | 1,110,124  | 689,759    |

Table 1: Open Images Dataset v6 statistics by data split.

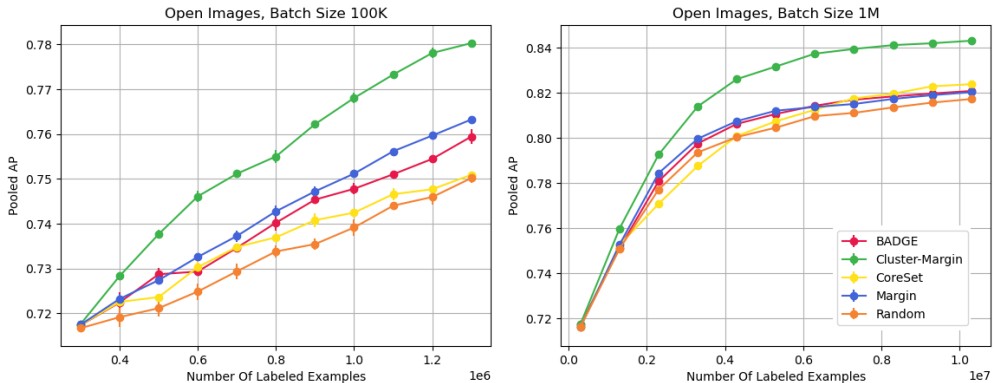

Figure 2: Pooled average precision of various active learning methods as a function of the number of labeled examples, using active learning batch sizes of 100K (left figure) and 1M (right figure). The mean and standard error as computed across three trials is shown. (The standard error bars are indeed barely visible.)

**Random Sampling:** selects $k$ examples uniformly at random from the set $X$. This baseline allows us to compare the benefit an active learning algorithm has over passive learning.

### 3.2 Open Images Dataset Experiments

We leverage the Open Images v6 image classification dataset [Krasin et al., 2017] to evaluate Cluster-Margin and other active learning methods in the very large batch-size setting, i.e. batch-sizes of 100K and 1M.

Open Images v6 is a multi-label dataset with 19,957 possible classes with partial annotations. That is, only a subset of these classes are annotated for a particular image (where on average, there are 6 classes annotated per image). The label for any given class is *binary*, i.e. positive or negative. Thus, if we have a positive label for image (img1) and class (dog) pair, this implies that there is a dog in the image img1. Similarly, a negative label on an image-class pair, (img2, dog), implies that a dog does not appear in the image, img2. Since in practice we need to decide which class to annotate for a given image, all active learning methods will be sampling image-class pairs and receiving a binary (positive/negative) label.

Table 1 lists the number of images as well as the number of positive and negative labels applied across images within the train, validation, and test folds. The training fold serves as the unlabeled pool which the active learning methods sample from. For a more detailed description of the dataset and distribution of labels across different data folds please visit the Open Images website.[2]

We train a ResNet-101 model implemented using tf-slim with batch SGD using 64 Cloud TPU v4's each with two cores. Each core is fed 48 examples per SGD iteration, resulting in an effective SGD batch of size $64 \times 2 \times 48 = 6144$. The SGD optimizer decays the learning rate logarithmically after every $5 \times 10^8$ examples and uses an initial learning rate of $10^{-4}$. We insert a final fully-connected hidden layer of 128 dimensions and use global pooling to induce a 128-dimensional feature embedding, which is needed by Cluster-Margin as well as the BADGE and CoreSet baselines.

We use a fine-tuning learning scenario in order to maximize training stability and reduce training variance: all trials are initialized with a model pre-trained on the validation split using 150K batch

---

[2]https://storage.googleapis.com/openimages/web/factsfigures.html

|  | BADGE | CoreSet | Margin | Random |
|---|---|---|---|---|
| Cluster-Margin 100K | 66% | 53% | 71% | 52% |
| Cluster-Margin 1M | 38% | 40% | 37% | 35% |

Table 2: The percentage of labels used by Cluster-Margin to achieve the highest pooled average precision of baselines on Open Images Dataset v6, for batch sizes of 100K and 1M.

SGD steps. Additionally, we sample a seed set of 300K image-class pairs uniformly at random from the unlabeled pool and train for an additional 15k steps. After this initialization phase, the active learning methods then sample a fixed number of image-class pairs (we consider both 100K and 1M) from the unlabeled pool at each active learning iteration. This additional sample augments the set of image-class pairs that have been collected up to that point and the model is then fine-tuned for an additional 15k steps using this augmented training set. We run 3 trials with a different random seed set for each method, with 10 active learning iterations per trial.

Recall, Cluster-Margin clusters the unlabeled pool as an initialization step at the beginning of the active learning process. In this case, HAC is run over the pool of images using feature embeddings extracted from the pre-trained model. We run a single-machine multi-threaded implementation [Sumengen et al., 2021] and, in all cases, we set $k_m = 10k_t$. We select $\epsilon$ such that the average cluster size of $\mathcal{C}$ is at least 10, allowing us to exhaust all clusters in the round-robin sampling. This allows Cluster-Margin to naturally sample images, but as discussed previously we want to sample image-class pairs. Thus, the sampling of $k$ pairs is computed in two steps: first Cluster-Margin samples $k$ images, then given all the potential classes with labels available in this set of $k$ images (recall, on average there will be $6k$ for this dataset), we sample $k$ image-class pairs uniformly at random.

The baseline methods also have to be modified for the multi-label setting. CoreSet follows a similar process as Cluster-Margin, where it first samples $k$ images, then it samples $k$ image-class pairs uniformly as random. Margin Sampling samples according to margin scores per image-class pair by using a binary classification probability per individual class. Similarly, BADGE calculates gradients per image-class pairs (that is, the gradient is composed of $l=2$ blocks), and runs the $k$-MEANS++ seeding algorithm on these image-class pair gradients.

BADGE and CoreSet runtimes are $O(dnk)$ where $d$ is the dimension, $k$ is the batch size and $n$ is the size of the unlabeled pool. For the Open Images dataset, $n \approx 9M$, $d = 256$, and $k$ is either 100K or 1M and thus, in order to run BADGE and CoreSet efficiently on this dataset, we partitioned uniformly at random the unlabeled pool and ran separate instances of the algorithm in each partition with batch size $k/m$ where $m$ is the number of partitions.[3] For the batch size 100K setting, we used 20 partitions for BADGE, while CoreSet did not require any partitioning. For batch size 1M, we use 20 and 200 partitions for CoreSet and BADGE, respectively. These parameters were chosen to ensure each active learning iteration completed in less than 10 hours, for all active learning methods.

The mean and standard error across trials of the pooled average precision (AP) metric (see Dave et al. [2021] for details) for each method is shown in Figure 2, for both the 100K and 1M batch-size settings. As expected, all active learning methods provide some improvement over uniform random sampling. Among the other baselines, in our experiment Margin sampling outperforms both the CoreSet and BADGE algorithms, apart from the final iterations of the 1M batch-size setting. Finally, we find that the Cluster-Margin algorithm significantly outperforms all methods in this task. In the 100K batch-size setting, the Margin algorithm is the second best method and achieves a final pooled AP of more than 0.76 after training with 1.3M examples. The Cluster-Margin algorithm achieves this same pooled AP after receiving only ∼920K examples – a reduction of 29%. In the 1M batch-size setting, we see even more extreme savings over the next best sampling method, with a 60% reduction in labels required to achieve the same performance. The percentage of labels required by Cluster-Margin to reach the final pooled AP of other methods is summarized in Table 2.

Finally, note that Margin was run as a baseline to ablate the effect of diversifying the low-confidence examples via clustering. As can be seen in Figure 2 and Table 2, Cluster-Margin requires the labeling of only 37% (resp. 71%) of examples compared to Margin for a batch size of 1M (resp. 100K).

---

[3]It is possible that a highly efficient parallelized approximate $k$-MEANS++ and $k$-center implementation would allow us to avoid partitioning the BADGE and CoreSet baselines.

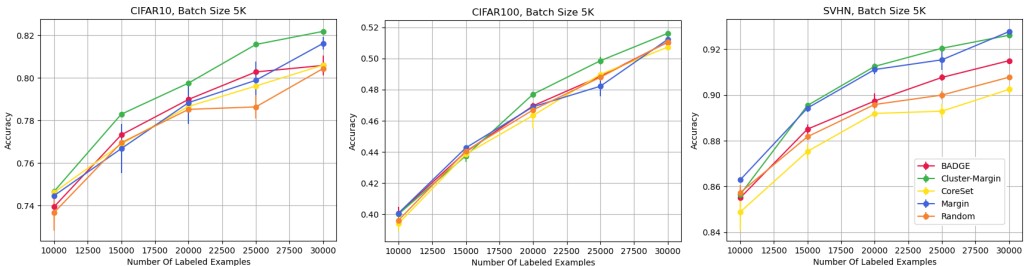

Figure 3: Accuracy of various active learning method as a function of the number of labeled examples, using active learning batch sizes of 5K. The mean and standard error as computed across 10 trials is shown.

### 3.3 CIFAR10, CIFAR100, SVHN Experiments

In this section, we compare our algorithm against the aforementioned baselines on three multi-class image datasets in the small batch-size setting. Although the focus of the paper is on the large scale setting, where we expect the largest headroom lies, the goal of these experiments is to verify that the proposed method performs well in smaller scale settings as well. Specifically, we consider CIFAR10, CIFAR100, and SVHN, which are datasets that contain 32-by-32 color images [Krizhevsky, 2009, Netzer et al., 2011]. For CIFAR10 and CIFAR100, the task is to classify the object in the image while for SVHN, the task is to classify street view house numbers. For CIFAR100, we used the 100 fine-grained labels. See Table 3 in the appendix for more details on these datasets.

We train a VGG-16 convolutional neural network model as implemented in the tf-keras library [Simonyan and Zisserman, 2015, Chollet et al., 2015]. We use batch SGD with learning rate fixed to 0.001 and SGD's batch size set to 100. We use the default pre-trained image-net weights to initialize the model. We add two fully-connected layers of 4096-dimension and prediction layer as the final layers of network. The embedding layer used for Cluster-Margin, BADGE and CoreSet are extracted from the penultimate layer of 4096 dimensions. In case of BADGE, we partition the data to improve efficiency on CIFAR100 (as discussed in the previous section); no partitioning was needed for CIFAR10 or SVHN. For all Cluster-Margin experiments, we set $k_m = 1.25k_t$, and set $\epsilon$ such that the average cluster size of $\mathcal{C}$ is at least 1.25, allowing us to exhaust all clusters in the round-robin sampling.

Since all datasets are multi-class, each active learning method will sample images (as opposed to image-class pair as was done in the previous section). Each active learning method is initialized with a seed set of size 10,000 which was sampled uniformly at random from $X$ and at each sampling iteration, the method will select 5,000 images. The sampling procedure is then repeated for 4 iterations. We repeat this entire experiment for ten trials and average the results.

Figure 3 shows that Cluster-Margin outperforms all baseline methods in terms of accuracy on CIFAR10 and CIFAR100 while both Cluster-Margin and Margin Sampling admit a similar performance on SVHN, that is above all other baselines (similar performance is seen with classification accuracy). BADGE attains a performance close to that of Margin Sampling on all datasets except on SVHN where Margin Sampling outperforms BADGE. Surprisingly, CoreSet does not go beyond the performance of Random Sampling on all datasets, which is perhaps due to our using of the 2-approximation for solving the $k$-center problem. In summary, even in the smaller scale setting, we find Cluster-Margin to be competitive with or even improve upon baselines.

## 4 Theoretical Motivation

We now provide an initial theoretical analysis to motivate the empirical success of Cluster-Margin. To this effect, we step through a related algorithm, which we call Cluster-MarginV, that is more amenable to theoretical analysis than Cluster-Margin, albeit less practical. We establish theoretical guarantees for the Cluster-MarginV algorithm, and show these guarantees also hold for the Cluster-Margin algorithm in specific settings, which will help shed some light on Cluster-Margin's functioning.

At a high level, the Cluster-MarginV Algorithm first samples uniformly along the margin of a hypothesis consistent with the collected data so far. Then, it selects from these points a diverse batch by leveraging a volume-based sampler that optimizes a notion of diameter of the current version space. If the volume-based sampler on the embedding space follows HAC-based sampling of the Cluster-Margin Algorithm, then Cluster-Margin and Cluster-MarginV are almost equivalent, other than the fact that Cluster-MarginV uniformly samples the data in the low margin region, instead of ranking all examples by margin scores and then extracting a diverse pool from them. Moreover, since the data in a deep neural embedding space tend to have a small effective dimension [Arora et al., 2019, Rahbar et al., 2019], and our active learning algorithms do in fact operate in the embedding space, we work out this connection in a low dimensional space.

The first sampling step of the Cluster-MarginV algorithm mimics that of the standard Margin Algorithm of Balcan et al. [2007]. Just as in the analysis in Balcan and Long [2013], we prove that Cluster-MarginV admits generalization guarantees under certain distributions. The Cluster-MarginV Algorithm admits label complexity bounds that improves over the Margin Algorithm by a factor $\beta$ that depends on the efficacy of the volume-based sampler, an improvement which is magnified when the data distribution is low dimensional.

After establishing this guarantee for general $\beta$, we give a specific example bound on the value of this factor $\beta$ of the form $\beta = d/\log(k)$, where $d$ is the dimensionality of the embedding space. This result holds for a particular hypothesis class and optimal volume-based sampler, and suggest that an improvement is possible when $d < \log k$, that is, when either the dimensionality $d$ of the embedding space is small or when the batch size $k$ is large, which is the leitmotif of this paper. We then show that this volume-based sampler is, in fact, approximately equivalent to the Cluster-Margin algorithm under certain distributions. We complement this result by also showing that $\log k$ is an upper bound on the improvement in query complexity for any sampler.

## 4.1 $\beta$-efficient Volume-Based Sampling and Connection to Cluster-Margin

We operate here with simple hypothesis spaces, like hyperplanes which, in our case, should be thought of as living in the neural embedding space.

Given an initial class $\mathcal{H}$ of hyperplanes, we denote by $V_i = \{w \in \mathcal{H} : \text{sign}(w \cdot x) = y, \ \forall (x, y) \in T\}$ the *version space* at iteration $i$, namely, the set of hyperplanes whose predictions are consistent with the labeled data, $T$, collected thus far. Given a labeled set $S$, we also define the closely related quantity $V_i(S) = \{w \in V_i : \text{sign}(w \cdot x) = y, \forall (x, y) \in S\}$. Further, let $d(w, w') = \Pr_{x \sim \mathcal{D}}[\text{sign}(w \cdot x) \neq \text{sign}(w' \cdot x)]$, and define $D_i(S) = \max_{w, w' \in V_i(S)} d(w, w')$, which can be thought of as the diameter of the set of hyperplanes in $V_i$ consistent with $S$.

**Definition 4.1.** *Let $S_i^u$ be a set of $k_{i+1}$ points that have been drawn uniformly at random from a given set $X$. We say that $\mathcal{V}$ is a $\beta$-efficient volume-based sampler, for $\beta \in (0, 1)$, if for every iteration $i$, $D_i(S_i^b) \leq \beta D_i(S_i^u)$, where $S_i^b$ is the set of $k_{i+1}$ points chosen by $\mathcal{V}$ in $X$.*

This definition essentially states that the sample selected by $\mathcal{V}$ shrinks the diameter of the version space by a factor $\beta$ more compared to that of a simple random sample.

The Cluster-MarginV Algorithm (pseudo-code in Algorithm 3 in Appendix B) at each iteration first selects a consistent hyperplane $\hat{w}_i$ over the labeled set $T$. It then selects uniformly at random a set $X_i$ of points from within the margin, so that $x \in X_i$ satisfies $|\hat{w}_i \cdot x| < b_i$. The algorithm then queries the labels of a subset $S_i^b \subset X_i$ of size $k_{i+1} \leq |X_i|/\gamma$ from $X_i$, where $\gamma > 1$ is a shrinkage factor for the the diversity enforcing subsampling. Recall that in our experiments with Cluster-Margin (Algorithm 2) on Open Images, we set this factor $\gamma$ to 10. The subset $S_i^b$ is selected here by a $\beta$-efficient volume-based sampler.

In Appendix B (Theorem B.1 therein) we show that replacing a uniform sampler within the low margin region (as is done by the standard Margin Algorithm of Balcan et al. [2007]) by a $\beta$-efficient volume-based sampler improves the label complexity by a factor $\beta$.

For a specific hypothesis class and volume sampler, we now prove a bound on $\beta$, and elucidate the connections to the Cluster-Margin algorithm under certain stylized distributions on the embedding space. All the proofs can be found in Appendix B.

**Theorem 4.2.** *Let $X \subset [0, 1]^d$ with distribution $\mathcal{D} = \otimes_{i=1}^d \mathcal{D}_i$ a product of $1$-dimensional distributions and $\mathcal{H} = \{\mathbb{1}_{x_i \leq v_i} | v \in [0, 1]^d\}$ be the set of indicator functions on rectangles with one*

*corner at the origin. Assume that $k = o(\sqrt{n})$. Let volume-based sampler $\mathcal{V}$ choose the points $S^b = \cup_{i=1}^d \{\arg\min_{x \in X} ||x - F_i^{-1}(jd/k)e_i||_2, j = 1, \ldots, k/d\}$ where $F_i$ denotes the cumulative distribution function of $\mathcal{D}_i$ and $e_i$ is the $i$th unit coordinate vector. Then $\mathcal{V}$ is a $\beta$-efficient volume-based sampler with $\beta = \frac{d}{\log(k)}$.*

This theorem implies that a volume based sampler operating on a low-dimensional embedding space may achieve a label complexity that is $d/\log(k)$ times smaller than that of the Margin Algorithm in Balcan and Long [2013], which can be substantial in practice, especially when $d$ is small and the batch size $k$ is large.

We connect this particular volume-based sampler to the Cluster-Margin algorithm of Section 2 in a specific setting by considering the simple case of $d = 1$ and uniformly distributed points. In this case, sampling according to the strategy in Theorem 4.2 is equivalent to creating $k$ clusters of equal sizes and choosing the center point from each cluster. This parallels the sampling of Cluster-Margin with distance threshold $\epsilon = 1/k$, which would create $k$ clusters of size at most $2\epsilon$ and sample a random point from each cluster, i.e., such a sample achieves $\beta = O(1/\log(k))$.

We note that, while this is a positive initial connection, equating volume based samplers and the Cluster-Margin algorithm more generally is an important open future direction.

We end this section by providing a general lower bound for $\beta$, which shows that the $1/\log(k)$ term in Theorem 4.2 cannot be improved in general.

**Theorem 4.3.** *Let $\mathcal{X} = \mathbb{R}^d$ and $\mathcal{H}$ be the set of hyperplanes in $\mathbb{R}^d$. Let $n = |X|$ and $k = o(\sqrt{n})$. Then there exists a distribution $\mathcal{D}$ on $\mathbb{R}^d$ such that if $X$ is sampled iid from $\mathcal{D}$, and $\mathcal{V}$ is any sampler choosing $k$ points, $D_i(S_i^b) = \Omega(1/\log k)D_i(S_i^u)$. Thus $\beta = \Omega(1/\log k)$ for all $\mathcal{V}$.*

## 5 Conclusion

In this paper we have introduced a large batch active learning algorithm, Cluster-Margin, for efficiently sampling very large batches of data to train big machine learning models. We have shown that the Cluster-Margin algorithm is highly effective when faced with batch sizes several orders of magnitude larger than those considered in the literature. We have also shown that the proposed method works well even for small batch settings commonly adopted in recent benchmarks. In addition, we have developed an initial theoretical analysis of our approach based on a volume-based sampling mechanism. Extending this theoretical analysis to more general settings is an important future direction.

**Acknowledgments.** We thank the anonymous NeurIPS reviewers whose comments helped us improve both the content and the presentation of this paper as well as the area chair for their careful handling of this paper.

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
