|          | Train  | Test   | # Classes |
|----------|--------|--------|-----------|
| CIFAR10  | 50,000 | 10,000 | 10        |
| CIFAR100 | 50,000 | 10,000 | 100       |
| SVHN     | 73,257 | 26,032 | 10        |

Table 3: CIFAR10, CIFAR100 and SVHN dataset statistics.

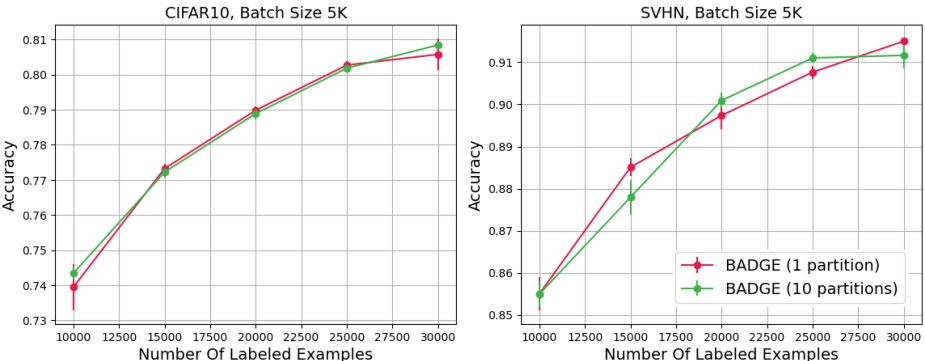

Figure 4: Accuracy of BADGE on CIFAR10 and SVHN when using one and ten partitions. The mean and standard error as computed across ten trials is shown.

# A    Extended Experiments

In this section, we expand on Section 3 by providing additional details and experimental results on the scalability of baseline methods and Cluster-Margin. Table 3 contains relevant statistics about the CIFAR10, CIFAR100 and SVHN datasets which have been omitted from the main body of the paper.

## A.1    Baseline Scalability

As discussed in Section 3, we improve BADGE's scalability on certain datasets by partitioning the unlabeled pool into subsets, and running BADGE independently on each subset. Specifically, if the size of the unlabeled pool is $n$, and $k$ is the batch size, we partition the pool uniformly at random into $m$ sets, and run BADGE independently with a target batch size of $k/m$ in each partition. The samples across all partitions are then combined. In order to measure the effect of partitioning, we run BADGE with both one partition and ten partitions on CIFAR10 and SVHN datasets. As can be seen in Figure 4, BADGE achieves comparable accuracy on both datasets. We do not include a plot for CIFAR100 because BADGE with one partition was not able to scale to this dataset. The same partitioning scheme was applied to CoreSet on only the Open Images dataset with 1M batch sizes. The Margin and Random baselines easily scale to all datasets and hence, the partitioning scheme was not used.

## A.2    Cluster-Margin Scalability

As described in Algorithm 2, while Cluster-Margin is more efficient than BADGE and CoreSet in each labeling iteration, it requires running HAC (Algorithm 1) on the entire pool, $X$, as a preprocessing step. Below, we discuss two independent techniques that can be used to speed up this preprocessing step.

**Multi-round HAC.** In order to avoid loading a complete graph into memory when running HAC, which can be infeasible for large datasets such as Open Images, we run a multi-round HAC approach in all of the experiments presented in Section 3. This approach involves clustering a set of clusters over multiple rounds, where in each round, HAC is run on the centroids of the clusters from the prior round. In each round, a $(k, \tau)$-nearest neighbor graph is built on the centroids of the clusters from the prior round, where an edge $(d_1, d_2)$ is created between two datapoints $d_1$ and $d_2$, if one datapoint is a $k$-nearest neighbor of the other, and the distance between $d_1$ and $d_2$ is at most $\tau$.

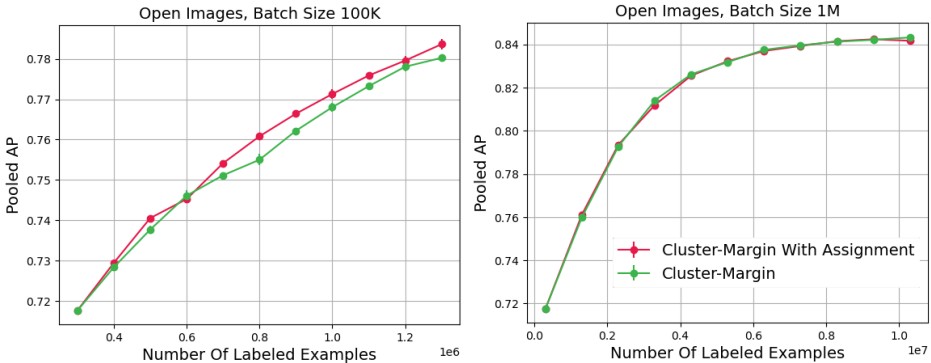

Figure 5: Pooled average precision of Cluster-Margin, as in Algorithm 2, compared to Cluster-Margin with the cluster assignment scalability improvement, using active learning batch sizes of 100K (left figure) and 1M (right figure). The green curves are identical to the ones in Figure 2. The mean and the (barely visible) standard error as computed across three trials is shown.

**Cluster assignment.** Another way to speed up Cluster-Margin is to run HAC on a smaller subset of the data during its pre-processing step. Thus, we execute another set of experiments where HAC is run *only* on the seed set image embeddings, $P$, which contains roughly 275K images (and 300K image-class pairs) and comprises only 3.1% of the entire set of images. We denote these clusters as $\mathcal{C}_P$. We then take the remaining 96.9% of image embeddings, and assign them to their nearest cluster centroid in $\mathcal{C}_P$, as long as the distance between the image and nearest centroid is at most $\epsilon_c$. We denote the final clusters as $\mathcal{C}'_P$. As was done in Section 3.2, we set $k_m = 10k_t$ for all experiments, and select threshold $\epsilon_c$ such that the average cluster size of $\mathcal{C}'_P$ is at least 10, allowing us to exhaust all clusters in the round-robin sampling. Figure 5 shows that on both batch sizes of 100K and 1M, the cluster assignment approach performs at least as well as the approach of running HAC on the entire set $X$, while reducing the clustering run-time from $O(|X|^2 \log |X|)$ to $O(|P|^2 \log |P| + |\mathcal{C}_P||X \setminus P|)$ where $|\mathcal{C}_P| \leq |P|$.

## A.3   Comparisons to Additional Baselines

We use this appendix to record additional evaluations conducted after the initial submission of this article.

In Figure 6 we consider the same setting as detailed in Section 3.3, but have additionally included the **FASS** algorithm of Wei et al. [2015]. This algorithm trades off uncertainty and diversity by first filtering away examples that the model is certain on (keeping $\beta k$ examples) and then greedily maximizing a submodular objective that encourages diversity. We run FASS with the $f_{\text{fac}}$ (facility location) objective and use $\beta = 1.25$, which we found to be a reasonable setting given the relative batch and pool sizes. In fact, using a much larger value of $\beta$ would imply that there is not much filtering done by the uncertainty sampling step. This choice also matches the factor used in Cluster-Margin. We find that the Cluster-Margin approach tends to outperform FASS on the CIFAR10 benchmark, while FASS performs comparably to Cluster-Margin on CIFAR100 and comparable to Cluster-Margin and simple Margin sampling in the SVHN benchmark.

From these additional experiments, we do not find that the overall conclusions we can draw out of this paper with respect to Cluster-Margin to have changed significantly.

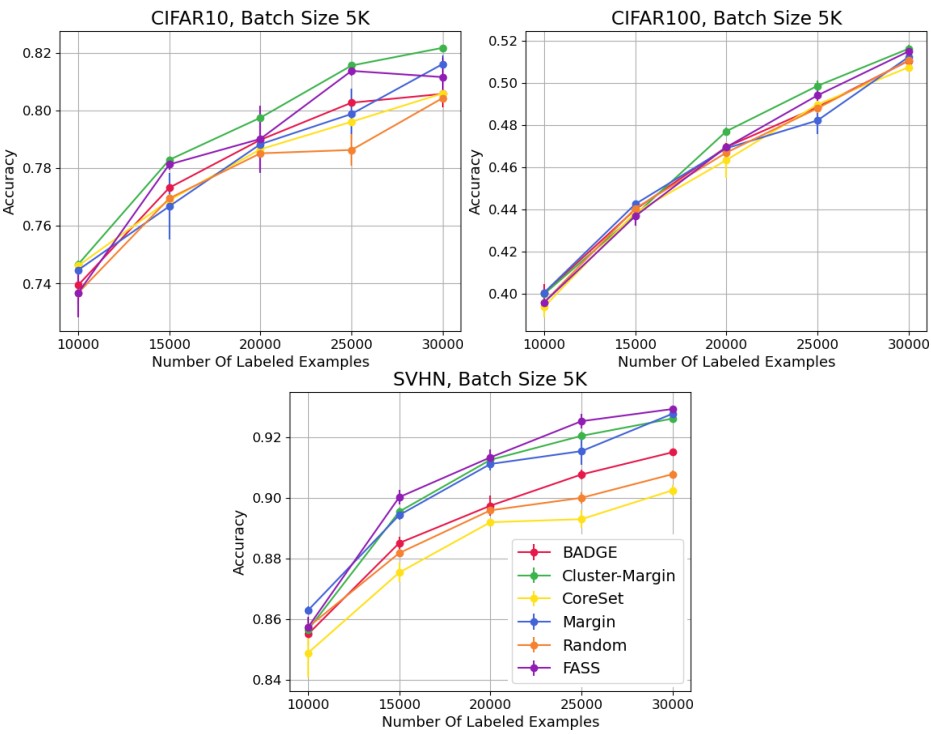

Figure 6: Accuracy of various active learning method as a function of the number of labeled examples, using active learning batch sizes of 5K. The mean and standard error as computed across 10 trials is shown.

## B    Extended Theoretical Motivation

Algorithm 3 contains a detailed pseudocode of the Cluster-MarginV algorithm referred to in the main body of the paper.

---

**Algorithm 3** Cluster-MarginV Algorithm

---

**Require:** (Unlabeled) distribution $\mathcal{D}$, $\epsilon$, $\delta$, volume-based sampler $\mathcal{V}$. Fix $\gamma > 1$.
1:  Set $c$, $C_1$, $C_2$, $n$ from Theorem B.1.
2:  $T \leftarrow \emptyset$, the set of labeled examples so far.
3:  **for** $i = 1, 2, \ldots, n$ **do**
4:      Set $b_i$, $k_{i+1}$ from Theorem B.1.
5:      Find hypothesis $\hat{w}_i$ ($\|\hat{w}_i\|_2 = 1$) consistent with $T$.
6:      $X_i = \emptyset$
7:      **while** $|X_i| < \gamma k_{i+1}$ **do**
8:          Sample $x \in \mathcal{D}$ and add to $X_i$ if $|\hat{w}_i \cdot x| < b_i$.
9:      **end while**
10:     Use $\mathcal{V}$ to select $S_{i+1}^b \subset X_i$ of size $k_{i+1}$.
11:     Query the labels of points in $S_{i+1}^b$.
12:     $T \leftarrow T \cup S_{i+1}^b$, update the set of labeled examples.
13: **end for**

---

The theorem that follows presents the generalization guarantee for the Cluster-MarginV Algorithm, as well as its label complexity guarantee. For simplicity, this result applies to a stylized scenario where labeled data is consistent with a given linear separator (the so-called realizable setting) and the data distribution in the embedding space is isotropic log concave.[4] Recall that a distribution over $\mathbb{R}^d$

---

[4]This result can be extended to a more general set of structured distributions by relying on the very recent results in Zhang and Li [2021], whose algorithm is a variant of the original Margin Algorithm that still samples uniformly from the low margin region, as Cluster-MarginV does.

is log-concave if $\log f(\cdot)$ is concave, where $f$ is its associated density function. The distribution is isotropic if its mean is the origin and its covariance matrix is the identity. Log-concave distributions are a broad family of distributions. For instance, the Gaussian, Logistic, and uniform distribution over any convex set are log-concave.

**Theorem B.1.** *Let the data be drawn uniformly from an isotropic log concave distribution $\mathcal{D}$ in $\mathbb{R}^d$ where $d \geq 4$ and $\mathcal{V}$ be a $\beta$-efficient volume-based sampler. Consider a class of homogeneous linear separators $w$ in the realizable setting and let $c$ be any constant such that for any two unit vectors $w, w' \in \mathbb{R}^d$, $c\theta(w, w') \leq d(w, w')$, where $\theta(w, w')$ is the angle between vectors $w$ and $w'$. Then, for any $\epsilon, \delta > 0$, $\epsilon < 1/4$, there exist constants $C_1, C_2$ such that the Cluster-MarginV Algorithm run with $b_i = \frac{C_1}{2^i}$ and $k_{i+1} = 2\beta C_2(d + \log \frac{1+n-i}{\delta})$ after $n = \lceil \log_2(\frac{1}{c\epsilon}) \rceil$ iterations returns with probability $1 - \delta$ a linear separator of error at most $\epsilon$. The label complexity is thus $O(\beta(d + \log(1/\delta) + \log\log(1/\epsilon))\log(1/\epsilon))$.*

*Proof.* The constant $c$ in the bound always exists by known properties of isotropic log concave distributions – see Lemma 3 in Balcan and Long [2013]. Define $R_i(S) = \max_{w \in V_i(S)} \Pr[\text{sign}(w \cdot x) \neq \text{sign}(w^* \cdot x)]$. Since $w^* \in V_i(S_i^b)$, it holds that $R_i(S_i^b) \leq D_i(S_i^b)$. Then by Definition 4.1 and the triangle inequality, it holds that

$$R_i(S_i^b) \leq D_i(S_i^b) \leq \beta D_i(S_i^u) \leq 2\beta R_i(S_i^u) \, . \tag{1}$$

We can analyze $R_i(S_i^u)$ by following a similar reasoning as in Theorem 5 of Balcan and Long [2013], except that we have to carefully deal with the $2\beta$ factor above in order to improve the overall bound. Concretely, let $P_1 = \{x \in \mathcal{X} : |w_{i-1} \cdot x| \leq b_{i-1}\}$ and $P_2$ be its complement. We split $R_i(S_i^u)$ into two components as follows

$$R_i(S_i^u) = \Pr[\text{sign}(w \cdot x) \neq \text{sign}(w^* \cdot x), x \in P_1] + \Pr[\text{sign}(w \cdot x) \neq \text{sign}(w^* \cdot x), x \in P_2] \, ,$$

and analyze each component separately.

For the $P_2$ component, Theorem 4 in Balcan and Long [2013] ensures that there exists a constant $C_1$ (which defines $b_i$) such that $\Pr[\text{sign}(w \cdot x) \neq \text{sign}(w^* \cdot x), x \in P_2] \leq \frac{c2^{-i}}{4\beta}$.

For the $P_1$ component, since $S_i^u$ is a uniform sample from region $P_1$, with $k_{i+1} = 2\beta C_2(d + \log\frac{1+n-i}{\delta})$, classic Vapnik-Chervonenkis bounds imply that $\Pr[\text{sign}(w \cdot x) \neq \text{sign}(w^* \cdot x)|x \in P_1] \leq \frac{c2^{-i}}{8\beta b_i}$. By Lemma 2 in Balcan and Long [2013], it holds that $\Pr[x \in P_1] \leq 2b_i$. Thus, $\Pr[\text{sign}(w \cdot x) \neq \text{sign}(w^* \cdot x), x \in P_1] \leq \frac{c2^{-i}}{4\beta}$.

Putting the two components together, the right-hand-side of Equation 1 is $c2^{-i}$, which concludes the proof. $\qquad\square$

Recall that under the same assumptions of the above theorem, the label complexity of the original Margin Algorithm is $O((d + \log(1/\delta) + \log\log(1/\epsilon))\log(1/\epsilon))$, as shown in Balcan and Long [2013]. Thus, the improvement of Cluster-MarginV over the Margin Algorithm is exactly by a $\beta$ factor in the label complexity.

## B.1 Proof of Theorem 4.2

We first establish a lemma that analyzes a one-dimension distribution setting.

**Lemma B.2.** *Let $X = \{x_i\}_{i=1}^n \subset [0, 1]$, $x_i$ i.i.d. with cumulative distribution function $F$, and $\mathcal{H} = \{\mathbb{1}_{x \geq v}(x) \mid v \in [0, 1]\}$ be the class of threshold-functions on $[0, 1]$. Assume that $k = o(\sqrt{n})$. Let sampler $\mathcal{V}$ choose $k$ points $S^b = \{s_i\}_{i=1}^k$ such that $s_i$ is the closest point in $X$ to the $i$th $k$-quantile of $F$: $s_i = \arg\min_{x \in X} |x - F^{-1}(i/k)|$. Then $\mathcal{V}$ is a $\beta$-efficient volume-based sampler for $\beta = O(1/\log k)$.*

*Proof.* Let $\mathbb{1}_u \in \mathcal{H}$ denote the threshold function with threshold $u \in [0, 1]$, and note that $d(\mathbb{1}_u, \mathbb{1}_v) = \Pr[x : \mathbb{1}_u(x) \neq \mathbb{1}_v(x)] = |F(v) - F(u)|$. Set $s_0 = 0$ and $s_{k+1} = 1$. Suppose the true hypothesis is

$\mathbb{1}_v$. Let $i$ be such that $s_i \leq v < s_{i+1}$. We have

$$\begin{aligned}
D(S^b) &= \max_i |F(s_{i+1}) - F(s_i)| \\
&= \max_i ((i+1)/k - i/k + O(1/\sqrt{n})) \\
&= \frac{1}{k}(1 + o(1)) \, .
\end{aligned}$$

On the other hand, suppose the $s_i$'s are chosen i.i.d. from $X$. Then $D(S^u) = \mathbb{E}[\max_i(y_{i+1} - y_i)]$, where $y_i = F(x_i)$. Now, the $y_i$'s are i.i.d. on $Unif([0,1])$ and it is known (Holst [1980]) that $\mathbb{E}[\max_i(y_{i+1} - y_i)] = \Theta(\log k/k)$. Thus $D(S^u) = \Theta(\log k/k)$ and we have $\beta = O(1/\log k)$. $\quad\square$

Now, we proceed to the proof of the theorem.

*Proof.* Knowing the labels of $S$ constrains the version space $V$ to be of the form $V = \{\mathbb{1}_{x_i \leq v_i} | v \in \prod_{i=1}^d [a_i, b_i]\}$. Furthermore, for each axis $i$, the interval $[a_i, b_i]$ is of the form $[F_i^{-1}(j/k), F_i^{-1}((j+d)/k)]$ for some $j \in \{0, 1, \ldots, k-1\}$. We claim that the diameter $D(S^b)$ of the hypothesis set $V$ given labels $S_i$ is bounded above by $D(S^b) \leq \frac{d^2}{k}$. To see this, first note that the pair of most distant hypotheses of $V$ are $\mathbb{1}_a := \mathbb{1}_{x_i \leq a_i}$ and $\mathbb{1}_b := \mathbb{1}_{x_i \leq b_i}$. Thus,

$$\begin{aligned}
D(S^b) &= P_{x \in \mathcal{D}}[1_a \neq 1_b] \\
&= \prod_i F_i(b_i) - \prod_i F_i(a_i) \\
&= \prod_i F_i(b_i) - \prod_i (F_i(b_i) - d/k) \\
&\leq (d/k) \sum_i F_i(b_i) \\
&\leq d^2/k.
\end{aligned}$$

Now suppose the $k$ points $T$ are chosen uniformly from $\mathcal{D}$. Fix an axis $i$, and project the sampled points onto this axis. Fix $\epsilon > 0$ and consider the $\epsilon k$ points (in expectation) that lie in the interval $[F_i^{-1}(1-\epsilon), 1]$. By the application of Holst [1980] in Lemma B.2 we can find an interval $[a_i, b_i] \subset [F_i^{-1}(1-\epsilon), 1]$ with $F_i(b_i) - F_i(a_i) \geq \epsilon(\log(\epsilon k)/(\epsilon k) = \log(\epsilon k)/k$ and such that none of the sampled points have $i$th coordinate falling in this interval. This means that the labels of $T$ are unable to distinguish between $\mathbb{1}_a$ and $\mathbb{1}_b$. The diameter $D(S^u)$ of the hypothesis set given labels of $T$ can thus be lower bounded by

$$\begin{aligned}
D(S^u) &\geq P_{x \in \mathcal{D}}[1_a \neq 1_b] \\
&= \prod_i F_i(b_i) - \prod_i F_i(a_i) \\
&= \prod_i (F_i(a_i) + \log(\epsilon k)/k) - \prod_i F_i(a_i) \\
&\geq (\log(\epsilon k/k) \sum_i F_i(a_i) \\
&\geq d \log(\epsilon k)/((1-\epsilon)k).
\end{aligned}$$

Taking $\epsilon = o(1)$, we have $\beta \leq D(S^b)/D(S^u) \leq (d/\log k)(1 + o(1))$. $\quad\square$

## B.2   Proof of Theorem 4.3

*Proof.* In the 1-dimensional setting, choosing the $k$ percentile points is optimal. This is because for any choice of $k$ points, $D_i(S_i^b)$ is determined by the maximal gap between successive points. Thus, the factor of $\beta = O(1/\log k)$ is sharp. Since we can embed the 1-dimensional setting in higher dimensions (by choosing a distribution that is supported on a 1-dimensional interval), we obtain a lower bound of $\beta = \Omega(1/\log k)$ in all dimensions. $\quad\square$