# OpenReview forum: "Batch Active Learning at Scale"
_NeurIPS.cc/2021/Conference — NeurIPS 2021 Poster_

### Official Review · Reviewer_8Xra · 2021-07-11

**Rating:** 6
**Confidence:** 3

**Summary:**

This paper proposes a large-batch active learning Clustering based algorithm, Cluster-Margin. The authors show that the algorithm is more efficient and more effective than comparable approaches (Random, Margin, CoreSet and BADGE) on several benchmark datasets (Open Images Dataset, CIFAR10, CIFAR100 and SVHN). The authors also provide initial theoretical guarantees justifying the relevance of the algorithm.


**Limitations And Societal Impact:**

It would be interesting to perform some experiments to assess if the results hold for other data modalities, like NLP (text) or ASR/Speech (audio) for instance.

**Main Review:**

Originality:
The algorithm seems to be novel and differs from previous comparable approaches like CoreSet or BADGE. Related work is adequately cited.

Quality:
The authors show empirically that their algorithm, Cluster-Margin, is both more efficient (O(nlog(n) vs O(n√(n)) than CoreSet and BADGE in practice and more effective. In particular, the algorithm clearly outperforms CoreSet, BADGE, Margin and Random on the Open Images dataset. The algorithm requires 29% less labels than the second-best model in the 100k batch-size setting and 60% less labels in the 1M batch-size setting to achieve the same result (mAP). Cluster-Margin also outperforms all other methods on CIFAR10, CIFAR100 and obtains a similar performance on SVHN.

The authors also establish a theoretical guarantee for the Cluster-MarginV algorithm and show that those results hold for the Cluster-Margin algorithm in specific settings. In particular, they show that the Cluster-MarginV algorithm has a label complexity bound which improves over the Margin algorithm by a factor beta. They also show that this improvement is possible, under specific hypotheses like an optimal volume-based sampler, when the dimensionality of the embedding space is small or when the batch size k is large. They also show that the optimal volume-based sampler is approximately equivalent to the Cluster-Margin algorithm. They finally show that log(k) is an upper bound on the improvement of query complexity for any sampler.
The authors are aware and mention that their theoretical results are initial and that equating volume based samplers and the Cluster-Margin algorithm is an open research question.

Clarity:
The paper is very clear and well organized. The authors detail the hyper-parameters and compute details used for the experiments. The Cluster-Margin algorithm is also explained in detail.

Significance:
The results are important as the algorithm allows for more efficient and effective large-batch-size active learning compared to existing methods. The authors also provide initial theoretical guarantees to explain the improvements obtained with the Cluster-Margin algorithm.



**Time Spent Reviewing:**

6

---

> ### Author Response · Authors · 2021-08-10
> **Response to Reviewer 8Xra**
>
> We thank the reviewer for the careful analysis of our work and feedback.
>
> > It would be interesting to perform some experiments to assess if the results hold for other data modalities, like NLP (text) or ASR/Speech (audio) for instance.
>
> In this work we focused on the commonly used vision applications in active learning literature to demonstrate the effectiveness of our proposed algorithm. We agree that evaluating on NLP and speech datasets will also be good and we plan to explore this in the future.

---

> > ### Comment · Reviewer_8Xra · 2021-09-06
> > **Re Rebuttal**
> >
> > I am satisfied with the authors' response. I keep my score the same (6).

---

### Official Review · Reviewer_m1qg · 2021-07-12

**Rating:** 8
**Confidence:** 4

**Summary:**

The paper introduces a new active learning acquisition function, Cluster-Margin, which clusters the samples with the lowest confidence scores (margin scores), and selects samples round-robin from the clusters to obtain a diverse set of low-confidence points. The experiments show that it works with very large batch sizes (100k--1M). Additionally, the paper provides an initial theoretical analysis of their algorithm.

**Limitations And Societal Impact:**

I agree with the author's answer in 1.c) about societal impact. Generally, the paper does not introduce concerns beyond the concerns of Active Learning itself, which are not part of the paper (but prior art): acquisition functions can introduce additional bias given how they select data to be labelled and included.

**Main Review:**

### Originality

The suggested method seems novel. Compared to the CoreSet algorithm in active learning, which computes a cover of all unlabeled pool points, Cluster-Margin essentially approximates a cover of mainly *low-confidence* pool points.

### Quality

The submission seems technically sound. The experiment results are impressive. Especially, the results on Open Images are impressive, but also the results on CIFAR-10 and CIFAR-100 show good improvements over the baselines. The Open Images results are great because a real-life dataset has other characteristics than smaller curated datasets ala CIFAR-10, etc, which shows promise for real-world applications of the proposed method.

### Clarity

The paper is very well written. The introduction and related work are great. The introduction is lacking citations, though. The algorithm and experiment section are also clear and of high quality. The theory section is difficult to understand and only seems to give a high-level overview of "Cluster-MarginV". The appendix is clearer on that.

I believe it might be worth extracting the theory section into a separate contribution. I believe the algorithm and results are sufficient on their own. The fact that the theoretical motivation comes at the very end seems telling.

### Significance

I assume the paper will be of significance for practitioners. An expanded theory section could be interesting overall. It would be nice to have more results on additional large datasets. On the other hand, the paper already provides results for the common three datasets (CIFAR-10, CIFAR-100, and SVHN), and then provides results for the large and high-dimensional Open Images Dataset. Within the constraints, this seems absolutely reasonable.

### Summary
Strengths:

+ method novel
+ very well written
+ experiments convincing: great results on large real-life datasets

Potential improvements
- the introduction might deserve more citations
- theoretical motivation needs to be more fleshed out (it feels a bit tacked on in its placement in the paper)

### Questions
1. Were there any experiments run that recompute the embeddings after every acquisition round? This could improve the performance as the quality of the embeddings ought to improve as more labelled data is used for training (esp given the large batch sizes).
2. Could Accuracy be provided instead of mAP for the CIFAR-10, CIFAR-100 and SVHN results? It is very hard to compare results as accuracy is the usual metric, and plots are not provided in the appendix either.

I would be happy to increase my score if the above questions are answered.

### Typos

Algorithm 2: line 5: $HAC(E_X, \epsilon, \infty)$: should the last argument be 0 instead of $\infty$? Otherwise, Algorithm 1 will early out right away in line 1.

**Time Spent Reviewing:**

6

---

> ### Author Response · Authors · 2021-08-10
> **Response to Reviewer m1qg**
>
> We thank the reviewer for their careful review and specific comments and questions. We answer them below:
>
> (1)
> This is indeed a natural idea, but it does come with the cost of reclustering the pool at each iteration. In initial explorations (not reported in the paper), we found that re-clustering does not provide enough improvement to outweigh the additional cost; we can clarify this in the paper.
>
> (2)
> Yes, in fact mAP is closely related to accuracy in the multi-class classification setting and the relative ordering and shape of the curves look identical to what is presented in Figure 3; we will update the figures in the paper to show accuracy, which, as you suggest may be more natural. We list the mean accuracies (and standard error) over 10 trials in the tables below:
>
> ### CIFAR10
>
> |Num Train|	10000|	15000|	20000|	25000|	30000|
> |-|-|-|-|-|-|
> BADGE|		0.739+/-0.009|0.773+/-0.002|0.790+/-0.001|0.803+/-0.002|0.806+/-0.006|
> Cluster-Margin|0.747+/-0.002|0.783+/-0.001|0.798+/-0.002|0.816+/-0.002|0.822+/-0.001|
> CoreSet|	0.746+/-0.002|0.769+/-0.002|0.787+/-0.001|0.796+/-0.002|0.806+/-0.002|
> Margin|	0.745+/-0.001|0.767+/-0.016|0.788+/-0.014|0.799+/-0.013|0.816+/-0.004|
> Random|	0.737+/-0.012|0.770+/-0.002|0.785+/-0.001|0.786+/-0.008|0.804+/-0.001
>
> ### CIFAR100
>
> |Num Train|     10000|  15000|  20000|  25000|  30000|
> |-|-|-|-|-|-|
> |BADGE|         0.400+/-0.006|0.440+/-0.006|0.470+/-0.002|0.488+/-0.004|0.510+/-0.002|
> |Cluster-Margin|0.400+/-0.002|0.437+/-0.005|0.477+/-0.002|0.499+/-0.003|0.516+/-0.003|
> |CoreSet|       0.394+/-0.007|0.439+/-0.004|0.463+/-0.012|0.490+/-0.002|0.507+/-0.002|
> |Margin|        0.401+/-0.002|0.443+/-0.003|0.469+/-0.004|0.482+/-0.009|0.512+/-0.005|
> |Random|        0.396+/-0.006|0.441+/-0.003|0.467+/-0.003|0.488+/-0.003|0.511+/-0.002|
>
> ### SVHN
>
> |Num Train|     10000|  15000|  20000|  25000|  30000|
> |-|-|-|-|-|-|
> |BADGE|         0.855+/-0.006|0.885+/-0.003|0.897+/-0.005|0.908+/-0.002|0.915+/-0.001|
> |Cluster-Margin|0.856+/-0.004|0.895+/-0.002|0.912+/-0.002|0.920+/-0.002|0.926+/-0.001|
> |CoreSet|       0.849+/-0.011|0.875+/-0.004|0.892+/-0.002|0.893+/-0.004|0.902+/-0.002|
> |Margin|        0.863+/-0.002|0.894+/-0.002|0.911+/-0.003|0.915+/-0.006|0.928+/-0.002|
> |Random|        0.857+/-0.005|0.882+/-0.002|0.896+/-0.001|0.900+/-0.003|0.908+/-0.002|
>
>
> Recall, these accuracies reflect the performance after training on a fraction of these benchmark datasets (at most 30k examples) and may not be entirely comparable with performance when training on the full dataset.
>
> > Should the last argument of HAC be 0 instead of $\infty$?
>
> Yes the last argument should be 0. Thank you for the correction.

---

> > ### Comment · Reviewer_m1qg · 2021-08-18
> > **Re Rebuttal**
> >
> > Re (1): Yes, please clarify.
> >
> > Re (2): Thank you. Could you also provide accuracies for the Open Images experiment, too?
> >
> > ---
> >
> > I'm satisfied with the reply and the additional provded information which I requested. I'm increasing my score to 8 (**conditioned on that you provide accuracies for Open Images, too**).
> >
> > Given that the AL experiments on CIFAR-10 etc were run on VGG-16 and without data augmentation (**could you clarify that explicitly?**), the accuracies for random and BADGE check out and are what I would expect. I assume no data augmentation because the text does not say that there was data augmentation.
> >
> > Thanks

---

> > > ### Author Response · Authors · 2021-08-23
> > > **Accuracy for Open Images**
> > >
> > > You are correct, all experiments were run without data augmentation.
> > >
> > > We have computed the average 0/1 accuracy across all classes and report the mean value of this measurement over the three trials in the table below. The standard error computed over the three trials was measured to be less than 0.003 for all values.
> > >
> > >  ### Open Images 1M
> > >
> > > |Train|	300000|	1300000|	2300000|	3300000|	4300000|	5300000|	6300000|	7300000|	8300000|	9300000|	10300000|
> > > |-|-|-|-|-|-|-|-|-|-|-|-|
> > > |BADGE	|0.437|0.522|0.575|0.602|0.617|0.623|0.630|0.634|0.637|0.638|0.642|
> > > |ClusterMargin	|0.438|0.531|0.597|0.635|0.660|0.670|0.675|0.679|0.679|0.682|0.683|
> > > |CoreSet	|0.438|0.534|0.580|0.609|0.630|0.639|0.644|0.650|0.649|0.656|0.656|
> > > |Margin	|0.438|0.524|0.582|0.611|0.625|0.636|0.638|0.643|0.645|0.651|0.652|
> > > |Random	|0.438|0.535|0.586|0.612|0.618|0.625|0.631|0.632|0.635|0.640|0.643|
> > >
> > > ### Open Images 100K
> > > |Train|	300000|	400000|	500000|	600000|	700000|	800000|	900000|	1000000|	1100000|	1200000|	1300000|
> > > |-|-|-|-|-|-|-|-|-|-|-|-|
> > > |BADGE|	0.439|0.462|0.475|0.483|0.494|0.504|0.513|0.520|0.527|0.534|0.542|
> > > |ClusterMargin|	0.438|0.469|0.485|0.504|0.515|0.527|0.539|0.551|0.561|0.569|0.576|
> > > |CoreSet	|0.439|0.457|0.465|0.477|0.489|0.495|0.504|0.509|0.518|0.522|0.530|
> > > |Margin	|0.437|0.462|0.476|0.489|0.502|0.511|0.521|0.531|0.542|0.549|0.557|
> > > |Random	|0.438|0.456|0.465|0.479|0.486|0.494|0.506|0.513|0.521|0.525|0.537|

---

> > > > ### Comment · Reviewer_m1qg · 2021-08-31
> > > > **Re**
> > > >
> > > > Thank you for providing this additional table.
> > > >
> > > > I'm satisfied with the performance of your proposed method both in the large acquisition batch setting and the more conventional setup.
> > > >
> > > > Best wishes

---

### Official Review · Reviewer_YihA · 2021-07-15

**Rating:** 3
**Confidence:** 5

**Summary:**

The paper proposes a large-scale batch active learning approach.  "large-scale" means large-scale batches, e.g., 100K-1M. The key idea is to first perform Hierarchical Agglomerative Clustering on the pool of data points. Each time, it compute the confidences of each data point (based on the difference between the largest two predicted class probabilities), and picks the ones with the lowest confidence, matches their clusters, and then samples the data points across these clusters in a round-robin fashion. The clustering is based on the second last layer's neurons (given the input). Some experiments were conducted to confirm the effectiveness of the method.

**Ethics Review Area:**

["I don’t know"]

**Main Review:**

The paper proposes a very simple method to select samples for active learning. The method makes sense. However, several severe issues are here.

(1) Skeptical motivation. The paper claims it aims to select very large batches for active learning, like 100K-1M. That sounds violating the principle of active learning.  AL is intended to reduce the number of labels (and labeling cost) as much as possible. They key is to smartly select the training examples, so that we can use much less samples to achieve good performance (comparable to abundant data).  If you are able to offer so many training examples, why not just run batch learning on the large dataset?  Why do you use AL?

(2) Skeptical efficiency. The paper claims that their batch selection is very scalable. However, the most costly part --- the HAC step, takes a complexity of O(n^2 log(n)) , which is very expensive and not scalable at all. see line 145. Although the paper says the HAC can be accelerated in practice from multi-thread implementation. The complexity is still there, more than quadratic to the number of candidates. Although the selection step is fast, the overall efficiency is very limited, given that the pool can be orders of magnitude larger than a batch.

(3) The paper claims BatchBALD (line 86) is infeasible for large batch size in practice. I think this is a wrong claim. BatchBALD is greedy heuristics, and the point is added into the batch one by one. It scales linearly in the size of the batch and data pool. The paper needs to explain why it is infeasible for large batch sizes.

(4) The paper mentions BatchBALD in Intro, but does not compare with it in the experiment. It misses an important soa as the baseline.

**Time Spent Reviewing:**

2

---

> ### Comment · Reviewer_m1qg · 2021-08-04
> **Re BatchBALD and AL setting**
>
> Re (3), (4):
>
> a. BatchBALD requires a BNN to draw correlated predictions from. Thus, it is not directly applicable to non-BNN models, like a ResNet-101 or VGG-16 without modifications. The paper seems to focus on non-BNN methods, which is generally common in active learning and compares to several SOTA methods that do not require BNNs, and which themselves compared favorably to BNN methods on publication.
>
> b. BatchBALD cannot scale easily to large acquisition batch size in the published form as it would require an incredibly large # of MC dropout samples to estimate the joint mutual information precisely enough to be of value. Mutual information is notoriously hard to estimate. The BatchBALD paper itself acknowledges the noisy estimator as a problem and states: "The quality of larger acquisition batches would be improved by reducing this noise."
>
> Hence, at the stated acquisition batch sizes, BatchBALD would surely not be able to perform better than uniform. Indeed, this is the case in my experience.
>
> Re (1):
>
> The reviewer might be aware that that are increasingly large unlabelled datasets being created by crawling the web, for example.
>
> While acquisition batch sizes of 100K-1M sound "mind-blowing", it does totally make sense to have methods for such large acquisition batch sizes in industrial settings, where you train on huge datasets and repeated retraining is (prohibitively) expensive (GPT-3 scale?). This is not missing the point of AL per-se, and the authors mention this motivation in their introduction.
>
> Best wishes
>
> PS: I would recommend spending more time reviewing than the stated 2 hours, especially when giving such a negative score with such high confidence.

---

> > ### Comment · Reviewer_YihA · 2021-08-04
> > **please look at the comments: odd motivation and problelmatic methods**
> >
> > I don't think this setting makes any sense. The goal of this paper contradicts much with the motivation of AL --- training with minimum labeling cost. If you are able to provide large samples, AL is not necessary.  I do NOT accept papers with odd motivations. The method is NOT novel yet problematic, and the paper over-claims its efficiency. see my comments --- I do not only question the motivation, but also the method and experiments.
> >
> >  " BatchBALD would require an incredibly large # of MC dropout samples" --- this is wrong. 100 samples are more than enough. MI is computed as an unbiased estimate --- according to the law of large numbers, the convergence is only determined by the number of samples,  not  by the dimension of the parameter space. More important, no matter how you criticize BatchBALD, you need to compare with it; you need to give the analysis, not just draw an arbitrary conclusion.
> >
> > PS: you don't need to doubt my reviewing time. I am an expert in this area.  2 hours are more than enough. I don't need to increase hour# or write lengthy/empty comments to stress on the confidence. again,  look at the comments --- they cover every aspect of the paper.

---

> > > ### Comment · Reviewer_m1qg · 2021-08-04
> > > **Re**
> > >
> > > BatchBALD does not work for such acquisition batch sizes. Indeed, we have found that with 100 MC samples, beyond acquisition size 15 or so, the BatchBALD scores become essentially uniform on MNIST, thus not being more informative than random acquisition. We have run experiments on this.
> > >
> > > Reflecting on it, the issue might also be the number of MC samples for computing the joint entropy of the model predictions. For acquisition batch size B with C classes, you need to approximate a sum over C**B terms with MC samples (the BatchBALD paper uses 10_000 MC samples).
> > >
> > > Also, while not being an expert in MC integration/summation, the MC error seems to be $V \frac{\sigma_n}{N}$, where $V$ is the domain's volume, which here would be $C^B$, so with each additional acquisition batch element, the error increases by a factor of C=10, which would explain the issue. I haven't thought about all this in a while, so apologies if my anecdotal evidence and quotes from the paper itself are stronger than the explanations I can provide here.
> > >
> > > I would encourage you to pursue an extension of BatchBALD for larger acquisition batch sizes if you think that is easily feasible, but that is unrelated to this paper.
> > >
> > > The published version of BatchBALD does not work for larger acquisition sizes, and this paper is not the first one to state this.
> > >
> > > The other fact also remains that BatchBALD is not applicable to non-BNNs, which is another limiting factor.
> > >
> > > Lastly, again, your argument about AL does not make sense. It is entirely possible to have very large unlabelled datasets and wanting to acquire a large number of labels at each acquisition step. If you have $10^9$ unlabelled samples, $10^6$ labels per acquisition step are not much.
> > >
> > > I also honestly don't believe you are an expert in this area contrary to your claims.
> > >
> > > Best wishes

---

> > > > ### Comment · Reviewer_YihA · 2021-08-04
> > > > **Re**
> > > >
> > > > 1. "BatchBALD does not work for such acquisition batch sizes. Indeed, we have found that with 100 MC samples, beyond acquisition size 15 or so, the BatchBALD scores become essentially uniform on MNIST, thus not being more informative than random acquisition. We have run experiments on this. "
> > > >
> > > > It is funny that you claimed BatchBALD does not work only after I questioned your one-sentence conclusion in the paper. If so, why didn't you report these results in the paper to support your conclusion? That will make your argument much more convincing. My review is based on the submitted paper. I am not a fun of BatchBALD. The point is why didn't you compare with BatchBALD in your paper?? It is a basic job, not an unreasonable requirement. It is weird  that you criticized an SOA in the intro (without any solid proof) yet not confirm  that SOA is indeed worse in practice.
> > > >
> > > > 2. Regarding MC complexity.
> > > >
> > > > Why don't you look at the BatchBALD paper (https://arxiv.org/pdf/1906.08158.pdf)? see. Sec 3.4, their complexity is linear  to BC. O(C^B) is the naive version. Please let me know their paper is wrong. I am happy to learn it.
> > > >
> > > > 3. The other fact also remains that BatchBALD is not applicable to non-BNNs, which is another limiting factor.
> > > >
> > > > Dropout training can be applied in any NN. That means, BatchBALD can be applied in those NN examples you listed before. If you want to view the inference results from a Bayesian perspective, you can get uncertainty estimate, etc. If you do not want, it is just an ensemble technique. How you view dropout does NOT determine if it is applicable to the actual NN.
> > > >
> > > > 4. meaningless of the large batch size.
> > > >
> > > > Well, just a simple question. If it is so easy to label 1M samples at once, why do you want to run 100 active learning, to train a model with 100M samples? Why not just collect 100M samples from beginning and train only once? The latter is ~100 times faster than AL. Then what is the meaning of AL? They key advantage of AL over batch learning is that it's faster in training (due to samples) and low cost of labeling. I don't see this work reflects these key properties of AL.
> > > >
> > > > 5. I also honestly don't believe you are an expert in this area contrary to your claims.
> > > >
> > > > LOL. Whatever you believe, I don't care.
> > > >
> > > > Finally, regarding my comment (2) --- skeptical efficiency --- I do not see any response. That's another major reason I do not believe the proposed method is truly scalable, even  let's put aside if the huge batch size in AL is useful or not.

---

> > > > > ### Comment · Reviewer_m1qg · 2021-08-04
> > > > > **I am a reviewer---not one of the authors**
> > > > >
> > > > > FYI, you are not arguing with the authors of the papers but with one of the reviewers.
> > > > >
> > > > > I just felt the need to correct several of your misleading statements regarding BatchBALD and Active Learning given that I am well acquainted with the matters we are discussing.
> > > > >
> > > > > As a reviewer, I can exactly see who I am talking to, and I invite you to do the same if you feel so necessary and to check my publication record as I have yours before I wrote my statement above. Hence, I do not believe you to be an expert in the wider area of active learning contrary to your claim, and neither am I.
> > > > >
> > > > > Re 1) As such, the information provided is from my own knowledge and experience with BatchBALD and from reading other literature which has found that claim to be true. I have noted your dislike for BatchBALD, but then I wonder even more why you are so adamant about it.
> > > > >
> > > > > Re 2) Regarding complexity, yes, and $C^B$ is what exactly when $C=10$ and $B \ge 10^5$? How would that be feasible for BatchBALD? Similarly, using BatchBALD's approximations, 3.4 and C, in the paper, would clearly not help or solve the issue.
> > > > >
> > > > > Re 3) This is a bad argument. You cannot simply add dropout to any architecture. It requires additional fine-tuning of dropout parameters if you want to do that correctly. Moreover, it is well-known that actual Deep Ensemble methods outperform MC dropout in active learning (see e.g. "The power of ensembles for active learning in image classification"). However, again BatchBALD does not scale that (even acquisition batch size 1000) and BALD suffers from redundancy, so lately there have been fewer comparisons. That is an issue for separate research but the onus is always on the Bayesians to show that the extra complexity is worth it. Active learning research clearly separates between Bayesian methods and non-Bayesian methods and non-Bayesian SOTA methods have consistently outperformed the Bayesian approaches in recent publications (at least in their experiments).
> > > > >
> > > > > Re 4) Clearly, AL is not about acquiring lots of unlabelled data, but it is about labelling cost as you have said yourself. The rest of your argument does not make sense. We can collect 100M samples by crawling the internet. Labelling each sample is a different beast. We could ask 1_000_000 users to help label certain samples using Captchas, which might take a few weeks to get reliable labels and then retrain. Retraining might cost millions of dollars of compute. It would definitely not be worth retraining more often given the cost.
> > > > >
> > > > > Now, I really don't understand how multiplying everything by a factor of 1000 changes the equation because we do write AL research papers that have acquisition sizes of 1_000 labels for datasets with 60_000 unlabelled samples.
> > > > >
> > > > > Regarding your comment (2), I will await the author's response. I took issue with (1), (3), and (4) in your review as the claims are questionable independently of this paper.
> > > > >
> > > > > Best wishes

---

> > > > > > ### Comment · Reviewer_YihA · 2021-08-04
> > > > > > **Re: can you do me a favor to understand my points before questioning --- I cannot waste time on meaningless discussions**
> > > > > >
> > > > > > "As a reviewer, I can exactly see who I am talking to, and I invite you to do the same if you feel so necessary and to check my publication record as I have yours before I wrote my statement above. Hence, I do not believe you to be an expert in the wider area of active learning contrary to your claim, and neither am I."
> > > > > >
> > > > > > You don't need to be humble. I never questioned if you are an expert in AL or not.  I am not interested to question that. If you intentionally look at my profile, it's far from complete. Again, I don't care if you view me as an AL expert or not. However, there is not any issue for me to understand this submission in 2 hours. To me, your argument to question my qualification is ridiculous.
> > > > > >
> > > > > > "Re 1) As such, the information provided is from my own knowledge and experience with BatchBALD and from reading other literature which has found that claim to be true. I have noted your dislike for BatchBALD, but then I wonder even more why you are so adamant about it."
> > > > > >
> > > > > > I told you that I am not a fun of BatchBALD.  The problem is the experimental design of the submitted paper, not the BatchBALD itself. Did you really read my sentences? I cited it here: "The point is why didn't you compare with BatchBALD in your paper?? It is a basic job, not an unreasonable requirement. It is weird that you criticized an SOA in the intro (without any solid proof) yet not confirm that SOA is indeed worse in practice" ---  You cannot defend for the authors by claiming that your experience is that BatchBALD is not working, worse than non-BNN method, etc, so that they can escape the experimentes. That's irresponsible. I need to see data or formal analysis in the paper to confirm their argument. If it lacks, the paper is problematic. Again, my review is based on the submitted paper. If you just  throw out a vague statement from your/others' experience, which I cannot check in any formal way, it CANNOT convince me.
> > > > > >
> > > > > > "Re 2) Regarding complexity, yes, and is what exactly when and ? How would that be feasible for BatchBALD? Similarly, using BatchBALD's approximations, 3.4 and C, in the paper, would clearly not help or solve the issue."
> > > > > >
> > > > > > Let me paste the time complexity here "O(bc min(c^b, m))". If the authors insist on that it is infeasible for large batches. Show me the analysis or practical running time in the paper. My concern is that the submitted paper claims BatchBALD does not work yet never shows any evidence (I don't think this is a common sense), even not compare the performance in small batches. That's NOT acceptable. If you want to help that paper, leave me the analysis AND suggest the authors to supplement the analysis in their paper. What you are doing is like you are trying to cover that paper's defect. That is extremely WEIRD.
> > > > > >
> > > > > > "Re 3) This is a bad argument. You cannot simply add dropout to any architecture. It requires additional fine-tuning of dropout parameters if you want to do that correctly. Moreover, it is well-known that actual Deep Ensemble methods outperform MC dropout in active learning (see e.g. "The power of ensembles for active learning in image classification"). However, again BatchBALD does not scale that (even acquisition batch size 1000) and BALD suffers from redundancy, so lately there have been fewer comparisons....."
> > > > > >
> > > > > > Why do you distort my words? Did I claim dropout can be NAIVELY applied to any NN? Did I claim dropout works better the deep ensemble? Let me cite my words again: "Dropout training can be applied in any NN. That means, BatchBALD can be applied in those NN examples you listed before." I said you can use dropout and batchbald to non-Bayes NNs for sure. The claimed advantage that "BatchBALD is not applicable to non-BNNs, which is another limiting factor" Is NOT correct.
> > > > > >
> > > > > >
> > > > > > "Re 4) Clearly, AL is not about acquiring lots of unlabelled data, but it is about labelling cost as you have said yourself. The rest of your argument does not make sense. We can collect 100M samples by crawling the internet. Labelling each sample is a different beast. We could ask 1_000_000 users to help label certain samples using Captchas, which might take a few weeks to get reliable labels and then retrain. Retraining might cost millions of dollars of compute. It would definitely not be worth retraining more often given the cost."
> > > > > >
> > > > > > Again, can you please seriously read my statement before you conclude it is "misleading"? My point is, if you are able to collect 1M samples to conduct AL for 100 steps. You have to train 100 times, and each training step is very costly. If you have had such amazing capability, why not collect 100M samples all together once (you already get big data), then just conduct one training of a giant model? That will much less costly. Plus, if you can really collect such big data, I don't believe smartly selecting or randomly selecting samples will make big differences. That's why I do NOT see AL is useful/better in this case.
> > > > > >
> > > > > > Please be sure to understand my points before questioning.

---

> > > > > > > ### Comment · Reviewer_m1qg · 2021-08-09
> > > > > > > **Citations**
> > > > > > >
> > > > > > > I only took a look because you claimed to be an expert to justify a 2-3-5 review that seems flawed and hurried imho.
> > > > > > >
> > > > > > > ### Re Active Learning & Large Acquisition Batch Sizes.
> > > > > > >
> > > > > > > Are you seriously claiming that you do not see the difference between obtaining unlabeled and labeled samples? It is sometimes easy to capture 1 billion unlabelled samples (crawl the internet, record frames from cars with video cameras). How do you figure out which of these samples to label if you only want to train with a few million of them at most? Does that sound like active learning?
> > > > > > >
> > > > > > > Anyhow, here is a citation, that this also looking at large scale AL with existing methods: ["Training Data Subset Search with Ensemble Active Learning" by Chitta et al, 2019](https://arxiv.org/abs/1905.12737). From the abstract:
> > > > > > > > our experiments on object detection are at the scale required for production-ready autonomous driving systems
> > > > > > > Maybe this paper can convince you of the use case?
> > > > > > >
> > > > > > > ### Re BatchBALD
> > > > > > >
> > > > > > > I went through a few papers that cite BatchBALD. Here are a few that discuss sampling issues with BatchBALD and might help bring the message home:
> > > > > > >
> > > > > > > ["Wat heb je gezegd? Detecting Out-of-Distribution Translations with Variational Transformers" by Xiao et al, 2019](http://bayesiandeeplearning.org/2019/papers/90.pdf):
> > > > > > >
> > > > > > > > Much worse, when attempting to naively use MI or entropy with long sequences or large sets of discrete random variables, we quickly discover that even approximate integration over the product space becomes
> > > > > > > prohibitive (Kirsch et al., 2019).
> > > > > > >
> > > > > > > ["Noisy Batch Active Learning with Deterministic Annealing" by Gupta et al, 2019](https://arxiv.org/pdf/1909.12473.pdf):
> > > > > > >
> > > > > > > > The batch version of BALD as proposed in [19]. The BBALD has difficulty in acquiring larger batches due to the exponential (Kb) computations needed or large Monte Carlo samples for sufficient accuracy.
> > > > > > > > [...]
> > > > > > > > The acquisition size is taken to be 100 to be able to use BBALD as well. We see in the Figure 11 that the proposed DeAn performs well in different noise strengths. We also observe that the recent BatchBALD perform inferior even to the BALD and Random. The reason being, computation of joint mutual information require O(Kb) computations which is exponential. The Monte-Carlo sampling used in the BatchBALD work approximate this term, the error of which grows with increase in K as well as b.
> > > > > > >
> > > > > > > I think this is quite clear and also provides experiments.
> > > > > > >
> > > > > > > Or somewhat related, ["Deep Multi-Fidelity Active Learning of High-dimensional Outputs" by Shibo Li, R. Kirby, Shandian Zhe, 2020](https://arxiv.org/pdf/2012.00901.pdf):
> > > > > > > > Third, we address the challenges in computing and optimizing the acquisition function. Due to the large output dimension, it is very expensive or even infeasible to estimate the required covariance and cross covariance matrices with popular Monte-Carlo (MC) Dropout (Gal and Ghahramani, 2016) samples.
> > > > > > >
> > > > > > > Lastly, and more recently, from a workshop I have attended:
> > > > > > >
> > > > > > > ["SIMILAR: Submodular Information Measures Based Active Learning In Realistic Scenarios", Kothawade et al, 2020](https://arxiv.org/pdf/2107.00717.pdf):
> > > > > > > > Similarly, BATCHBALD [25] does not scale to larger batch sizes since their method would need a large number of Monte Carlo dropout samples to obtain a significant mutual information. Such limitations reduce the scope of applying these methods to realistic settings.
> > > > > > >
> > > > > > > I think this sentence in the introduction that you seem to have such problems with is neither novel, unexpected, false, nor sufficient grounds to make up half of the reasons for your clear rejection. The paper compares plenty of methods that can scale to larger acquisition batch sizes without turning into uniform selection. It is sufficiently well known that BatchBALD will not scale to these acquisition batch sizes that further experiments to confirm the already known are not required.
> > > > > > >
> > > > > > > If you want to argue for clear rejection, I would suggest you come up with better reasons.
> > > > > > >
> > > > > > > Thanks and best wishes

---

> > > > > > > > ### Comment · Reviewer_YihA · 2021-08-17
> > > > > > > > **Re**
> > > > > > > >
> > > > > > > >
> > > > > > > > Thanks for these references. However, you do not address my concerns; **in fact, your arguments even have nothing to do with my concerns**
> > > > > > > >
> > > > > > > > 1. I've made this point very clearly. **You are NOT the authors. It is the authors' responsibility to use experiments, references and analysis to support their argument, NOT you**. That's what I asked for. I would be very happy to see the paper had included these. Unfortunately, they are lacking, and the current paper is in a very casual form, which is NOT acceptable. If you write your dissertation in this way, it is very unlikely you are gonna pass.
> > > > > > > >
> > > > > > > > 2. You did NOT address my concern about the motivation either. My point is in the training cost, not the sample selection. Again, please show some respect to understand my point before attacking. In fact, even for sample selection, I am skeptical about its meaning under large sample sizes.
> > > > > > > >
> > > > > > > > 3. let me give this back to you: **If you want to cover the defect of this paper, I would suggest you come up with better reasons.**

---

> > > > > > > > > ### Comment · Reviewer_m1qg · 2021-08-18
> > > > > > > > > **Re**
> > > > > > > > >
> > > > > > > > > I have replied plenty to your "concerns". And the authors have also replied with a meaningful rebuttal.
> > > > > > > > >
> > > > > > > > > The point in my response above was to show that what you have been asking for is unreasonable and tangential to the actual paper. I give clear evidence of that.
> > > > > > > > >
> > > > > > > > > I consider your review as made in bad faith---at best because you didn't want to put in the effort to actually review the paper and went looking for strawmen to motivate a rejection. Maybe, you should not have picked BatchBALD for that, but anyhow: you have only been doubling down on bad arguments ever since, and I have doubts about some of your concerns given your own prior art, which I will raise separately.
> > > > > > > > >
> > > > > > > > > You are contributing to why submitting to conferences has been a terrible experience lately. Most authors submitting to NeurIPS spend many hours on their papers, and they deserve better reviews. Given your position, one would also expect better.
> > > > > > > > >
> > > > > > > > > When you don't want to review a paper, just decline the review right away and ask for a different paper instead of sabotaging it. There is no harm in that.
> > > > > > > > >
> > > > > > > > > Best wishes

---

> > > > > > > > > > ### Comment · Reviewer_YihA · 2021-08-18
> > > > > > > > > > **Re**
> > > > > > > > > >
> > > > > > > > > > Wow, **you first attack my qualification, and then attack my review as "made in bad faith". You are making every effort, except responding outright to my concerns about the paper. It sounds like personal rather than professional.** Well, I am not a psychologist, and I don't know **what your underlying intention is and why you are working so hard to push a disqualified paper to get in**. I am not interested in that. This thread of discussion is meaningless, cause it doesn't address my questions. I'll keep the score and leave the decision to the area chair and program chair.

---

> ### Author Response · Authors · 2021-08-10
> **Response to Reviewer YihA**
>
> We thank the reviewer for their specific comments and answer them, as numbered, below:
>
> (1)
> Even in the large scale setting, labeling costs are a bottleneck. For example, in our experiments in Sec 3.2, we give a budget to sample ~10M labels, which is just a fraction of the full 57M labels available in the pool. Figure 2 demonstrates that the proposed sampling method provides significantly more model improvement per label and, thus, is a more efficient use of our labeling budget.
>
> Another way to interpret Figure 2 (right panel) is that Cluster-Margin sampling produces a model that outperforms of all other sampling methods (i.e. achieves an AP > 0.825) after receiving only ~4M labels, i.e. a more than 2x savings in labeling cost compared to these other baselines. This again represents a huge practical savings in the large scale setting.
>
> Regarding the large batch size: in the large scale setting, frequently re-training the model is prohibitively costly, which restricts the number of rounds of labeling that we can conduct -- this naturally leads to the large batch sizes that this work is investigating. In a very large-scale multi-label experiment, as presented in the paper, one should by no means be led to the conclusion that the difference between “smartly selecting” and “randomly selecting” is minimal. If that were the case, we would not see the performance difference we report in Figure 2 (compare, e.g., Cluster-Margin to Random therein).
>
> (2)
> In section A.2 of the supplementary material we provide two methods (Multi-Round HAC and Cluster Assignment) that we use to improve the efficiency of the HAC step. Most significantly, in the ‘Cluster assignment’ subsection, we demonstrate experimentally that $n$ can be decreased dramatically, without the loss of quality. In particular, we propose a strategy of running HAC on only the seed set, $P$, to generate clusters $\mathcal{C}_P$, and projecting the remaining points $X \setminus P$ onto $\mathcal{C}_P$. This reduces the run-time from $O(n^2 \log n)$ to $O(|P|^2 \log |P| + |\mathcal{C}_P||X \setminus P|)$ where $|\mathcal{C}_P| \leq |P|$. On Open Images, this translates to running HAC on only 3.1% of the entire dataset. As shown in Figure 5, the quality of Cluster-Margin is maintained. We will move this discussion and experiments to the main body of the paper in the revised version; thank you for pointing this out and our apologies for not pointing to this discussion previously.
>
> (3) & (4)
> In our experience (using code posted by the authors https://github.com/BlackHC/BatchBALD), we found the algorithm to be an order of magnitude slower than the other baselines we compared to. Although the algorithm incrementally selects elements of a batch in a greedy fashion, the computational cost (Monte-Carlo samplings) required to score the remaining pool after each incremental addition becomes significant. In particular, using the provided default modest setting of 10 MC dropout samples, we found that sampling a single batch of 500 points from the MNIST dataset takes 20 hours using 6 cores of a 3.6GHz Intel Xeon E5-1650. The smallest batch size we use in our experiments is 5,000, which would roughly translate to 200 hours per active learning iteration (1000 hours for a single full trial). The running time on the three small datasets of Section 3.3 would be in the same ballpark as above. This is indeed too slow to be practical in this setting.
> We will add this discussion in the revised version.

---

### Comment · Area_Chair_GRKf · 2021-08-25
**Additional emergency review**

Based on the discussions between reviewers m1qg and YihA, I have decided to obtain an additional emergency review for this paper, which can be found below. It would be great if the authors could comment on this review.

Emergency reviewer:

"Generally, I would think of it as a borderline accept paper (score 6), since it is a very simple strategy that is easy to implement, which could be a useful baseline for batch AL for future research.  The experimental setting seems impressive, in the sense that the batch size considered in this paper (100k-1M regime) is quite large compared to most previous works.  Last but not the least, the author provided a somewhat related theoretical analysis through a variant of the margin algorithm.

However, there are also a few weaknesses. For example, the author chose the margin score as the measurement for predictive uncertainty, which is less justified given there are so many other metrics available. Also, the paper only considered BADGE and CoreSet as baselines. However, there are also other popular methods in the literature that could be considered in the batch AL setting. For example, the FASS [1] algorithm and, and more recently the Glister algorithm [2] both incorporates uncertainty and diversity at the same time. Finally, although the authors claimed that BALD and BatchBALD can not be applied to large batch sizes, I do think that they can perfectly handle datasets like MNIST (especially their improved variants such as [3], which could cope with a batch size of ~10^3 on CIFAR 10). It would be helpful to see more comparisons with those classic methods (or their recent variants).

References

[1] Wei, Kai, Rishabh Iyer, and Jeff Bilmes. "Submodularity in data subset selection and active learning." International Conference on Machine Learning. PMLR, 2015.

[2] Killamsetty, Krishnateja, et al. "Glister: Generalization based data subset selection for efficient and robust learning." arXiv preprint arXiv:2012.10630 (2020).

[3] Kirsch, Andreas, Sebastian Farquhar, and Yarin Gal. "A Simple Baseline for Batch Active Learning with Stochastic Acquisition Functions." arXiv preprint arXiv:2106.12059 (2021)."

---

> ### Author Response · Authors · 2021-08-31
> **Response to additional review**
>
> We thank the additional reviewer for their valuable comments.
>
>
> Below, we provide a discussion of the additional algorithms and an empirical comparison with the FASS algorithm [1] on the smaller datasets (Cifar10, Cifar100 and SVHN). Since [2] is a generalization of [1], this will also cover in part the algorithmic ideas in [2]. The limited amount of time we have during this discussion does not allow us to run comprehensive experiments against all algorithms mentioned in the new review. We will do our best to conduct similar experiments with the Open Images dataset in the final version of the paper, assuming we can successfully scale the FASS algorithm to batches of size 100K-1M.
>
>
> The FASS algorithm is closer in spirit to our ClusterMargin algorithm in that it balances informativeness and uncertainty. As mentioned in [1], it relies on diversity enforcing mechanisms whose principled theoretical motivation is derived from Nearest-Neighbor and Naive Bayes models, and may not necessarily match well with DNNs. Diversity is induced by approximately optimizing a submodular function. Each evaluation of this function, even after leveraging a sparse nearest-neighbor graph, may result in $O(n)$ running time. Notice this holds irrespective of whether, for example, greedy, lazy greedy, or randomized versions thereof are leveraged. To our knowledge, the best of these optimization procedures will require at least $\Omega(k)$ function evaluations, resulting in a total runtime of $O(nk)$, which in the large batch setting is prohibitively large (e.g., $k$ = 1M and $n$ = 10M). This is a potentially more expensive per-batch computation than the ClusterMargin approach, which only scores and sorts candidates once for each batch construction ($O(n \log n)$ cost). Reference [3] contains a simple idea to turn the original BALD algorithm into a batch active learning algorithm via a softmax function over the current uncertainties in the pool. This idea yields a faster and simpler algorithm than BatchBALD, but it does not appear to explicitly model diversity (other than the sampling without replacement). We note that the paper was posted on arXiv only two months ago, which is after our submission, and it was presented to an ICML workshop only last month.
>
>
> All in all, the references mentioned are relevant to this paper, and we will certainly cite and discuss them in the revised version.
>
>
> Below we share the empirical results we obtained on CIFAR10, CIFAR100, and SVHN with the addition of the FASS algorithm. Using the terminology found in [1], we run FASS with the $f_{{\mbox{fac}}}$ (facility location) objective and use $\beta$=1.25, which we found to be a reasonable setting given the relative batch and pool sizes. In fact, using a much larger value of $\beta$ would imply that there is not much filtering done by the uncertainty sampling step. We find that the ClusterMargin approach tends to outperform FASS on the CIFAR10 benchmark, while FASS performs comparably to ClusterMargin on CIFAR100 and comparable to ClustonMargin and simple Margin sampling in the SVHN benchmark.
>
> To summarize, we do not find that the conclusions we can draw out of this particular empirical analysis with respect to ClusterMargin to have changed significantly. Apart from this, as mentioned previously, we do not yet have a complete picture of the scaling properties of FASS when applied to the large batch sizes involved in learning with very big datasets like Open Images (the leitmotif of our paper).
>
>
>
> ### CIFAR 10
>
> |Num Train|	10000|	15000|	20000|	25000|	30000|
> |-|-|-|-|-|-|
> |BADGE|0.739±0.009|0.773±0.002|0.790±0.001|0.803±0.002|0.806±0.006|
> |ClusterMargin|0.747±0.002|0.783±0.001|0.798±0.002|0.816±0.002|0.822±0.001|
> |CoreSet|0.746±0.002|0.769±0.002|0.787±0.001|0.796±0.002|0.806±0.002|
> |FASS| 0.737±0.012|0.781±0.002|0.790±0.016|0.814±0.002|0.812±0.010|
> |Margin|0.745±0.001|0.767±0.016|0.788±0.014|0.799±0.013|0.816±0.004|
> |Random|0.737±0.012|0.770±0.002|0.785±0.001|0.786±0.008|0.804±0.001|
>
>
>
> ### CIFAR 100
>
> |Num Train|10000|15000|20000|25000|	30000|
> |-|-|-|-|-|-|
> |BADGE|0.400±0.006|0.440±0.006|0.470±0.002|0.488±0.004|0.510±0.002|
> |ClusterMargin|0.400±0.002|0.437±0.005|0.477±0.002|0.499±0.003|0.516±0.003|
> |CoreSet|0.394±0.007|0.439±0.004|0.463±0.012|0.490±0.002|0.507±0.002|
> |FASS|0.396±0.006|0.437±0.007|0.470±0.003|0.494±0.004|0.515±0.002|
> |Margin|0.401±0.002|0.443±0.003|0.469±0.004|0.482±0.009|0.512±0.005|
> |Random|0.396±0.006|0.441±0.003|0.467±0.003|0.488±0.003|0.511±0.002|
>
>
>
> ### SVHN
>
> |Num Train|10000|15000|20000|25000|	30000|
> |-|-|-|-|-|-|
> |BADGE|0.855±0.006|0.885±0.003|0.897±0.005|0.908±0.002|0.915±0.001|
> |ClusterMargin|0.856±0.004|0.895±0.002|0.912±0.002|0.920±0.002|0.926±0.001|
> |CoreSet|0.849±0.011|0.875±0.004|0.892±0.002|0.893±0.004|0.902±0.002|
> |FASS| 0.857±0.005|0.900±0.003|0.913±0.004|0.925±0.003|0.929±0.002|
> |Margin|0.863±0.002|0.894±0.002|0.911±0.003|0.915±0.006|0.928±0.002|
> |Random|0.857±0.005|0.882±0.002|0.896±0.001|0.900±0.003|0.908±0.002|

---

> > ### Comment · Area_Chair_GRKf · 2021-09-01
> > **Comment by emergency reviewer**
> >
> > The response addresses some of my issues and the additional experiment is helpful. I would keep my score.

---

### Decision · Program_Chairs · 2021-09-27

**Decision:**

Accept (Poster)

**Comment:**

A majority of reviewers voted for acceptance (including the ones shown as official reviewers and an emergency reviewer that I included last minute). The only reviewer voting for rejection is reviewer YihA, which seems to be too negative without strong justifications given that most of his claims are not supported by the other reviewers. I, therefore, have decided to accept the paper.